# A computational framework to study sub-cellular RNA localization

Aubin Samacoits[1,2], Racha Chouaib[3,4], Adham Safieddine[3,4], Abdel-Meneem Traboulsi[3,4], Wei Ouyang [1,2], Christophe Zimmer [1,2], Marion Peter[3,4], Edouard Bertrand[3,4], Thomas Walter [5,6,7] & Florian Mueller [1,2]

RNA localization is a crucial process for cellular function and can be quantitatively studied by single molecule FISH (smFISH). Here, we present an integrated analysis framework to analyze sub-cellular RNA localization. Using simulated images, we design and validate a set of features describing different RNA localization patterns including polarized distribution, accumulation in cell extensions or foci, at the cell membrane or nuclear envelope. These features are largely invariant to RNA levels, work in multiple cell lines, and can measure localization strength in perturbation experiments. Most importantly, they allow classification by supervised and unsupervised learning at unprecedented accuracy. We successfully validate our approach on representative experimental data. This analysis reveals a surprisingly high degree of localization heterogeneity at the single cell level, indicating a dynamic and plastic nature of RNA localization.

[1] Unité Imagerie et Modélisation, Institut Pasteur and CNRS UMR 3691, 28 rue du Docteur Roux, 75015 Paris, France. [2] C3BI, USR 3756 IP CNRS, 28 rue du Docteur Roux, 75015 Paris, France. [3] Institut de Génétique Moléculaire de Montpellier, University of Montpellier, CNRS, Montpellier, France. [4] Equipe labellisée Ligue Nationale Contre le Cancer, Paris, France. [5] MINES ParisTech, PSL-Research University, CBIO-Centre for Computational Biology, 75006 Paris, France. [6] Institut Curie, PSL Research University, 75005 Paris, France. [7] INSERM, U900, 75005 Paris, France. Correspondence and requests for materials should be addressed to E.B. (email: edouard.bertrand@igmm.cnrs.fr) or to T.W. (email: thomas.walter@mines-paristech.fr) or to F.M. (email: muellerf.research@gmail.com)

Non-random sub-cellular RNA localization is important for cellular function and its misregulation is linked to a number of diseases[1,2]. Initially observed in highly polarized cells such as oocytes or embryonic fibroblasts, more recent studies revealed diverse and wide-spread RNA localization in other systems[3], including bacteria[4], yeast[5], and developing embryos of fruitfly, ascidians and zebrafish[3,6]. RNA localization also occurs in cultured mammalian cell[7–9]. Besides the particular case of neurons where a large number of mRNAs localize in cellular processes, mRNA localization also occurs in regular cell lines to regulate gene expression at the spatial level. Secreted and mitochondrial proteins are often translated at the endoplasmic reticulum and mitochondria, respectively, while mRNA repressed for translation can accumulate in P-bodies or stress granules. More specific examples of localization include mRNAs that accumulate at the tip of cellular extensions[9], localize at the cell periphery[10], or DYNC1H1 mRNA that accumulates in foci representing dedicated translation factories[11]. With the rapid development of high-throughput techniques, it is likely that many more localized RNAs will be discovered. However, validated analysis tools to identify and classify such RNA localization patterns are currently lacking.

Imaging technologies, especially single molecule FISH[7,12,13] (smFISH), allow to observe single RNA molecules in their native cellular environment. This technique is now easy to implement and can be performed at low cost[13]. It provides unique quantitative spatial information[2,7] and thanks to recent advances, can be performed at large scale in cell lines and embryos[7,10,12,14,15]. Image analysis then allows to discover genes displaying non-random localization patterns. While many localization patterns are distinguishable by visual inspection[3,8], manual annotation can be biased, is often not quantitative and influenced by confounding factors such as RNA expression level. In addition, comprehensive manual annotation at the single cell level hardly seems an option for larger scale studies where thousands of cells are imaged in a single experiment. Indeed, the benefits of automatic analysis of smFISH data[7,16] include scalability and reproducibility, allowing for an accurate and quantitative description of the spatial aspects of gene expression.

In smFISH images, individual RNA molecules appear as bright diffraction-limited spots, which can be localized in 3D with published image analysis tools[12,14]. In contrast to the analysis of cellular phenotypes[17] and protein localization[18], smFISH data can be treated as point clouds. The smFISH signal inside a cell can thus be represented by features describing this spatial distribution of points, such as the mean nearest neighbor distance between spots or their average distance to the nuclear envelope. These features can then be used to group cells based on similarity in their RNA localization patterns, using supervised or unsupervised machine learning methods[7]. However, one of the main difficulty in this approach is the absence of a ground truth for RNA localization in smFISH data, making it impossible to assess usefulness of features and performance of the classification workflow. Hence, as of today, there is no rigorously validated method to analyze smFISH data at the cellular level.

Here, we present a simulation framework to create a synthetic ground-truth data set to perform this validation. Such simulated ground-truth data provide a number of key advantages to the traditional strategy relying exclusively on manual annotation[17–21]. Manual annotation of 3D point clouds irrespective of their number and reference volume is time consuming, difficult, error prone and tends to be subjective, in particular for subtle differences. In addition, we can only annotate already observed patterns from already identified example genes. This encouraged us to build a simulation framework in order to complement or replace manual annotation. We generated point patterns from known localization rules to create large amounts of ground-truth data. This allowed us to also control the parameters of the generative model in order to study robustness and limitations of the automatic algorithms. We show that the simulation of a large set of images enables designing and validating workflows for unsupervised and supervised analysis of smFISH data, which are capable of detecting a large variety of localization classes. We applied this approach to experimental data and successfully detected the different manually annotated localization classes. We also implemented a metric to quantitatively analyze heterogeneity of localization patterns. Application of this metric to our experimental data set revealed a surprisingly high degree of localization heterogeneity.

## Results

**Simulating realistic ground-truth data**. We designed a simulation framework capable of generating ground-truth data closely mimicking experimental smFISH images. First, we acquired fluorescence microscopy data to determine the 3D volume of cells and nuclei, as well as the typical smFISH background signal. We then placed individual RNAs with realistic signals according to pre-defined localization rules inside these cellular volumes. This set of simulated images allowed us to validate the entire analysis workflow from RNA detection to the identification of localization classes. Our approach to generate such synthetic smFISH images consists of four main steps (Supplementary Note 1):

First, we inferred accurate 3D cellular shapes from experimental data[22]. To do so, we performed an smFISH experiment in HeLa cells using Cy5-labeled probes against the highly expressed *GAPDH* mRNA. We defined the 3D cellular outline as the conforming boundary containing all detected mRNAs (Fig. 1a, b). We further used our observation that *GAPDH* mRNA was largely absent from the nucleus (Fig. 1a) to determine the average height and position of nuclei in the cells (Fig. 1b). This provided us with a collection of cellular and nuclear volumes, where RNAs can be placed. For each cell, we also acquired another channel with realistic background from a mock smFISH experiment, using Cy3-labeled probes against a not-expressed reporter gene. In addition, we acquired images for 2D segmentation of cells and nuclei (CellMask[TM] and DAPI, respectively), as they are used in standard screening applications. The open design of our workflow makes it possible to add additional marker channels, e.g., P-bodies, for more specific screening applications.

Second, we defined different RNA levels. It is well known that expression levels can vary greatly between genes and even between clonal cells for a single gene. Ongoing efforts identify stochastic noise in transcription and extrinsic factors such as cellular microenvironment or cell volume as the molecular and environmental origins of these cell-to-cell variations[22,23]. In contrast, an analysis workflow of RNA localization has to be independent of RNA levels, because different cells should be grouped together based on similarity in their RNA localization patterns, and not in their expression levels. We defined RNA density as a free parameter, and the absolute number of RNAs was assumed to be proportional to the cell volume, as shown in recent studies[22,23] (Supplementary Note 1). We simulated four regimes of expression, each regime with constant RNA density modulated with an additional Poisson noise term, which when pooled cover a large range of expressions levels in agreement with a recently observed large-scale screen (Fig. 1c)[7]. Lastly, we also considered a scenario with very high expression levels to test the potential limitations of the classification approaches.

Third, we simulated realistic images of individual RNA molecules. Each RNA was simulated as a point-spread-function (PSF) with sub-pixel localization and intensities from an

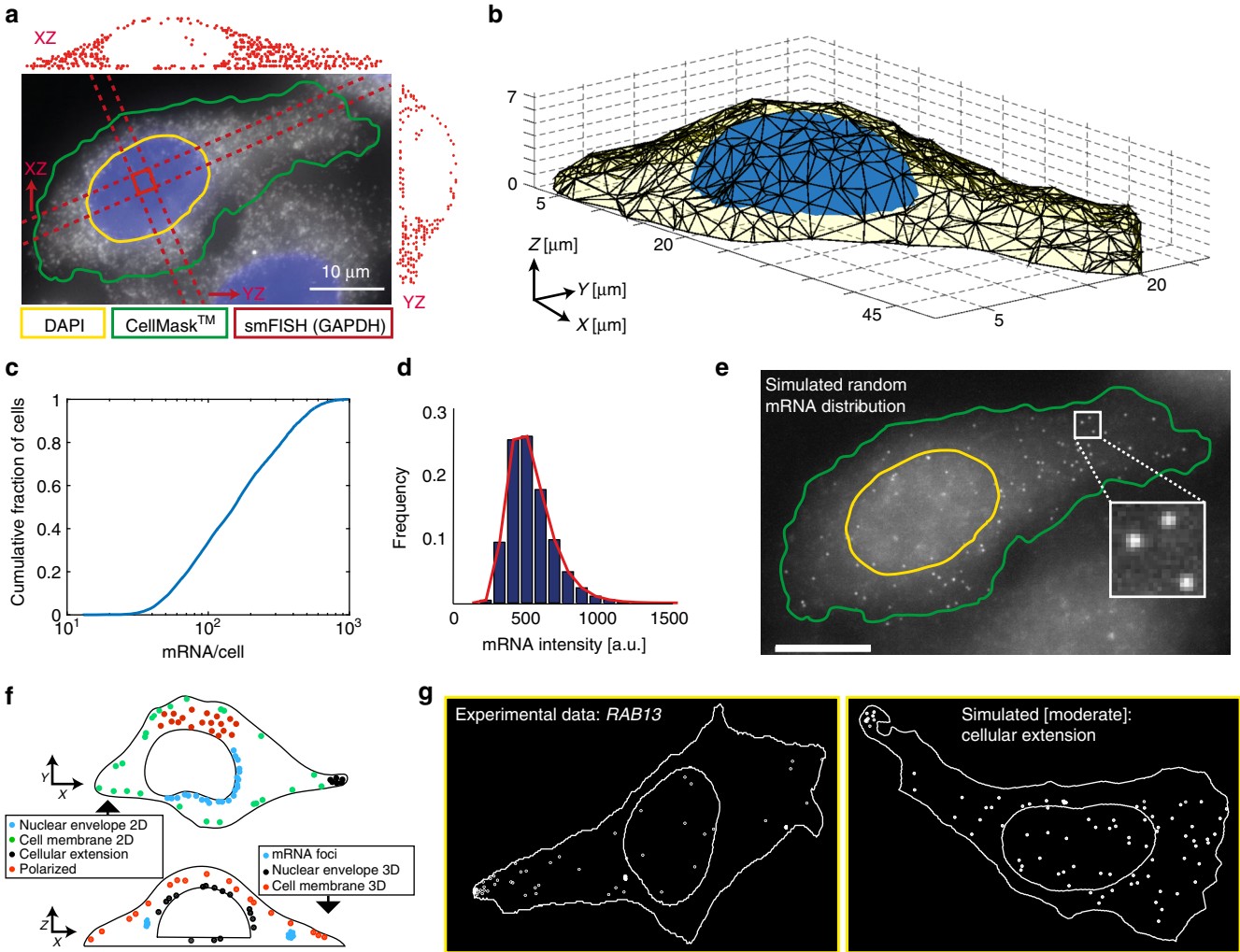

**Fig. 1** Simulation of smFISH images. **a** smFISH experiment against GAPDH (probes labeled with Cy5) in HeLa cells shown as a maximum intensity projection along Z (MIP). Cell segmentation (green) was performed with CellMask™, nuclear segmentation (yellow) with DAPI signal. Projections of detected GAPDH mRNAs located between the dashed lines are displayed as indicated by the arrows as side panels. **b** Final 3D polygon of a cell (yellow) and its nucleus (blue). **c** Cumulative histogram of pooled expression level. **d** mRNA intensity distribution extracted from experimental data (*KIF1C* mRNA). Histogram is fitted with a skewed normal distribution (red). **e** Example of a simulated cell with random mRNA localization shown as a MIP. Outline of cell and nucleus in green and yellow, respectively. Scale bars 10 μm. **f** Cartoon illustrating the simulated seven non-random localization patterns. **g** Outlines of a cell and nucleus with mRNA positions of the pattern cell extension: experimental data (*RAB13* mRNA) (left), simulated data with low mRNA density and moderate pattern strength (right)

experimentally measured distribution (Fig. 1d, Supplementary Note 1). Together with the realistic smFISH background, this simulation strategy also allows testing the performance of RNA detection algorithms [12,14] (Supplementary Note 2).

Fourth, we placed RNAs within the 3D cellular volume either randomly (Fig. 1e) or according to one of seven specific localization patterns (Fig. 1f–g). We designed these patterns to either mimic experimental data or to generate so far unobserved, yet plausible localizations. The patterns correspond to RNAs localizing to the tip of cellular extensions, around the cell membrane in 2D or 3D, with a polarized pattern, in foci, and around the nuclear envelope in 2D or 3D. Each pattern corresponds to a set of rules defining the spatial distribution of RNA molecules. For each pattern, there is a set of fixed parameters that define the nature of the localization pattern. In addition, we define one parameter (pattern strength) to control how extreme a pattern is (e.g., for the nuclear envelope localization, the pattern strength corresponds to the fraction of RNA in proximity of the envelope). For low pattern strength, the

distribution approaches spatial randomness. For more details we refer to Supplementary Note 1. For this study, we defined three strengths, with the intermediate corresponding to a typically observed experimental pattern as assessed by a human observer (for an example see Fig. 1g, where the localization of *RAB13* mRNAs in cellular extensions[9] is compared to simulations). The pattern strength can be used to study the sensitivity of the workflow in discriminating localized and non-localized RNAs. Taken together, we can simulate realistic images with different RNA expression levels and pattern strength.

**Analysis of simulated data with existing approaches.** We first generated data with little heterogeneity for both the pattern strength and expression level. We used simulated images with moderate pattern strength and an average of 200 RNAs per cell (100 cells per condition), to assess the performance of previously published analysis methods aiming at identifying RNA localization patterns. Our first analysis was based on a recent publication

presenting a very comprehensive analysis workflow[7,24]. Here, 32 localization features are calculated from a mRNA detection performed on a maximum-intensity projection of the smFISH images (Supplementary Note 4).

In order to visualize the feature distributions for the simulated patterns, we projected these features onto two dimensions by t-Distributed Stochastic Neighbor Embedding (t-SNE)[25]. While half of the patterns were reasonably well separated, the other half was inseparable (Fig. 2a). The inseparable patterns were RNA localizing in cellular extensions, in foci, at the cell membrane in 3D or randomly. k-means clustering yielded the same set of indistinguishable patterns (50% accuracy, Fig. 2b). This encouraged us to improve RNA detection in 3D and to design new features describing their spatial distribution.

**RNA detection and localization features**. Large-scale studies have shown that accumulation of RNA in cytoplasmic foci is a predominant localization pattern[3]. Despite their biological relevance, previous analysis approaches were not designed to correctly identify these foci. We hypothesized that this might be due to problems during RNA detection in existing methods, where an accumulation of very close RNAs is detected as a single molecule. We hence designed an analysis method based on Gaussian Mixture Models (GMM), where these foci are decomposed into individual RNA molecules. We detail the validation of this approach on simulations and experimental data in Supplementary Note 2.

We then designed new feature families based on established concepts from spatial statistics and image analysis. First, we used Ripley's L-function, which provides information about homogeneity of spatial density. We assumed that this would be particularly useful for detection of patterns like foci and polarized RNAs. Second, we used morphological operators to extract cellular extensions from the 2D mask of the cells and used the enrichment ratio of RNA counts in these extensions as a feature. Third, we developed features capturing RNA localization with respect to the cell membrane. This is challenging, since usually no information about the 3D cellular shape is available in standard smFISH experiments. By using our mock smFISH experiment, we could show that the estimated background of wide-field smFISH images was correlated with the cell height (Supplementary Note 3). We thus defined as a feature the correlation of the measured z-positions of RNAs and the background intensity (approximation for cell height). Lastly, we normalized the features describing distances between RNAs and cellular structures by the expectation of those distances under complete spatial randomness. Using our simulated data, we found that this normalization reduced the impact of cellular shape on the feature distributions, as compared to previously proposed normalization schemes[7] (Supplementary Note 3). We also added two previously published features for the description of polarized mRNA localization (Polarization and dispersion index)[16]. Altogether, we defined a new set of features (14 new and 9 previously published, Table 1).

We then analyzed the simulated data with this new feature set and the new RNA detection scheme in 3D including the GMM. This greatly improved the identification of localization patterns: all localization patterns were distinguished in an unsupervised setting with an overall accuracy of 90% (Fig. 2c, d, Supplementary Note 4).

**Robustness of classification toward heterogeneity**. We next investigated the robustness of this classification toward increased heterogeneity. First, we pooled the three pattern strengths defined above and found that the classification results remain good

(accuracy 79%; Supplement Note 4). We then pooled data simulated with RNA densities from an average of 50–400 RNAs per cell (Supplement Note 4), corresponding to the range recently measured in a large-scale smFISH study [7]. This test is crucial, since the analysis should identify cells with similar RNA localization patterns independently of variations in their expression level. We found that in a t-SNE plot, cells remained separated based on their localization pattern (Fig. 2e), showing that the features are stable with respect to expression heterogeneity. We also analyzed simulated cells with a very high expression level (average of 800 RNAs per cell). Here, we found that 7 out of 8 patterns could still be correctly identified (Supplementary Note 4). Only RNA foci were confounded with random, which can be explained by the increased local density of genes expressed at very high levels. However, we expect such extreme levels to be rare.

We then compared various clustering strategies (Supplementary Note 4) and found that k-means applied to a six-dimensional t-SNE analysis gave the best results (88% accuracy, Fig. 2f). If the number of clusters is not known, it can be inferred from the data with traditional methods such as the silhouette score or an analysis of the within-class variability (Supplementary Note 4).

Lastly, we tested whether our simulations could also be used to analyze the extent by which two spatial distributions must differ to be still detectable as being different. This is particularly interesting in the context of drug perturbation experiments. Specifically, we can use our simulation framework to perform a sensitivity analysis. We simulated data with a large range of pattern strength and investigate the impact of the pattern strength on the accuracy of the analysis (Supplementary Note 4).

Overall, we designed a workflow for the unsupervised analysis of localization patterns, that we validated on simulated data with varying degrees of both pattern strength and expression heterogeneity. Our results indicate that, while some localization patterns are easier to detect than others, all these localization patterns could be detected with good accuracy.

**Detection of localized mRNAs in experimental data**. Motivated by these results, we analyzed experimental data from 10 genes in HeLa cells (150–400 cells per gene, 2600 cells in total; Supplementary Note 5). Three mRNAs were manually annotated as random (*KIF20B*, *MYO18A*, *PAK2*), and seven were chosen because of their non-random localization. *RAB13*[9] and KIF1C mRNA accumulate in cell extensions. *DYNC1H1*[11] and BUB1 mRNA concentrate in cytoplasmic foci. The last three mRNAs display a localization associated with the nucleus: the *ATP6A2* protein is synthesized on the endoplasmic reticulum and its mRNA concentrate in the perinuclear region; *SPEN* mRNA form a rim that decorates the nuclear edge; while *CEP192* mRNA concentrates inside the nucleus. Some of these patterns were not explicitly included in our simulated classes (e.g., perinuclear and intranuclear), but they were nevertheless included to test whether our feature set would be general enough to enable classification of novel patterns. It is also important to note that, for any of these genes, we typically observe different cellular subpopulations with different spatial distributions of the encoded transcripts. Gene-level annotation therefore describes only a tendency and cannot be considered as a single-cell annotation.

We applied the workflow that we had benchmarked on the simulated data to the experimental data and extracted localization features for each cell. We represented the extracted localization features as a t-SNE plot (Fig. 3a, Supplementary Note 5). We also created a more detailed version of this plot where each point was replaced by a thumbnail representation of the cell, showing outlines of cells and nuclei and the detected RNAs. The plot can

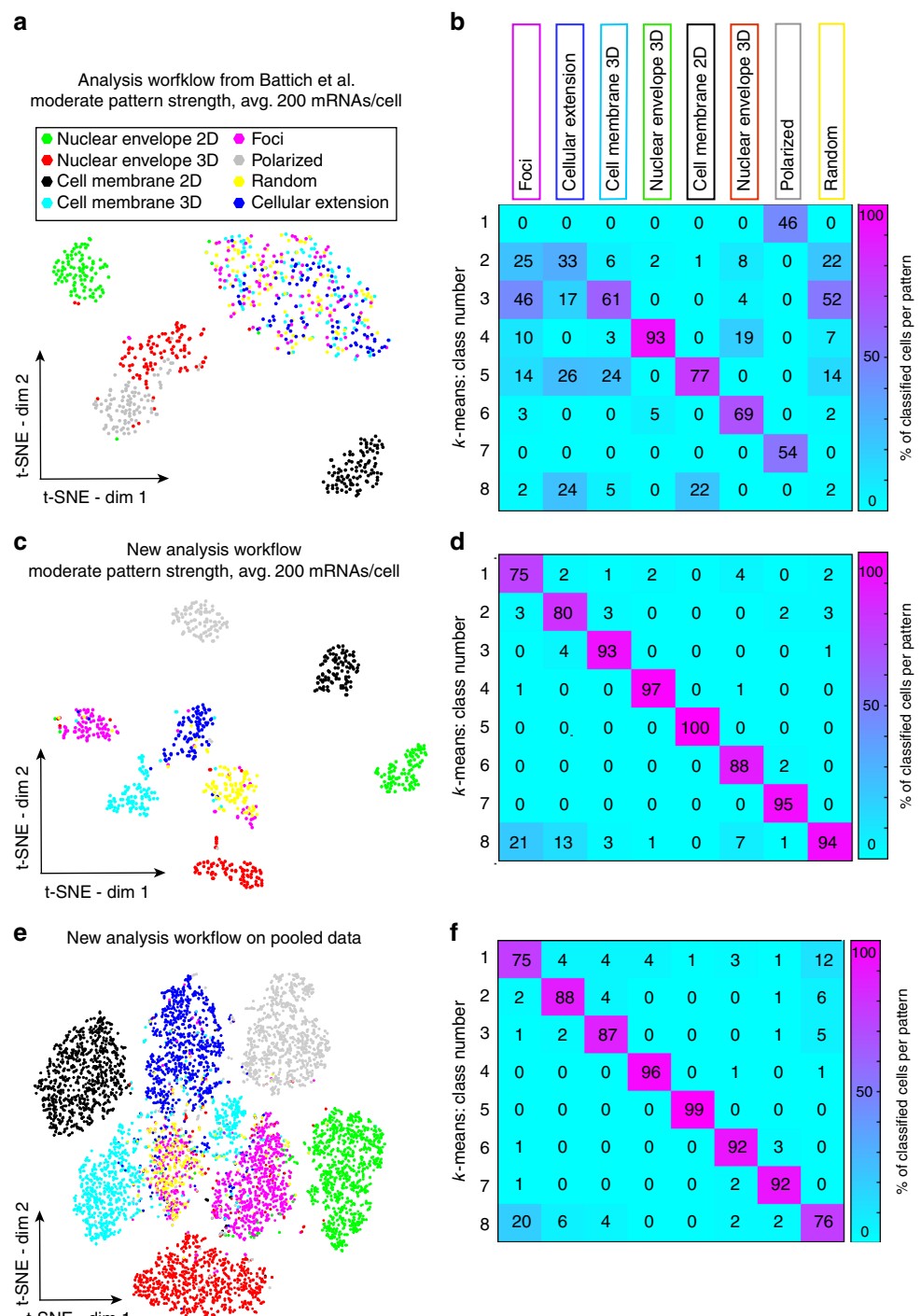

**Fig. 2** Analysis of simulated data. **a** Analysis with previously published mRNA detection approach and localization features. In the t-SNE plot, each dot is a cell colored with its localization pattern. **b** Confusion matrix for k-means clustering with 8 classes for data in **a**. Rows show identified classes, columns the known localization pattern. Numbers in each square indicate percentage of cells for a localization pattern (column) that was assigned to a given class (row). Off-diagonal elements are mis-classifications. **c**, **d** t-SNE projection and confusion matrix as in **a** and **b** but determined with our 3D spot detection with GMM and the new localization features. **e** Analysis of pooled simulated data (different pattern strength and expression levels). Analysis and t-SNE projection as in **c**. **f** Confusion matrix for data in **e** with k-means clustering performed on t-SNE projected data in six dimensions

be viewed at https://muellerflorian.github.io/locFISH_deepzoom/ #results/tsne_exp (for examples of the thumbnails see Fig. 3a). The tools to generate such plots are part of the provided Matlab source code. This zoomable t-SNE plot gives a clear picture of what the localization patterns in different regions of the feature space look like.

We found that cells from different genes with the same or similar manually annotated localization pattern showed an important overlap in the t-SNE plot. Conversely, we found that cells from genes with different manual annotations tended to populate distinct regions (Fig. 3a). This shows that the localization features are well suited to describe mRNA

**Table 1 List of localization features used in Figs. 3 and 4**

| Feature ID | Feature description |
| --- | --- |
| 1 | Ripley: maximum |
| 2 | Ripley: max gradient [0,max] |
| 3 | Ripley: min gradient [max,end] |
| 4 | Ripley: value at mid-point between center and boundary |
| 5 | Ripley: Spearman correlation between Ripley and radius |
| 6 | Ripley: radius of max value |
| 7 | Polarization index |
| 8 | Dispersion index |
| 9–12 | Morph opening–enrichment ratio: 15, 30, 45, 60 pixels |
| 13 | Cell height: Spearman correlation with $Z_{mRNA}$ |
| 14 | Cell height: $R^2$ with $Z_{mRNA}$ |
| 15 | Cell membrane: distance – mean |
| 16-19 | Distance membrane: quantile 5%, 10%, 20%, 50% |
| 20 | Nucleus: distance – mean |
| 21 | Cell centroid: distance – mean |
| 22 | Nucleus centroid: distance – mean |
| 23 | Ratio: mRNAs inside nucleus/outside nucleus |

localization from experimental data. This result was confirmed by the analysis of a different cell line (C2C12) (Supplementary Note 5). These cells are larger than HeLa cells (600 vs. 200 μm²) and also more elongated. We found that C2C12 cells with three different manually annotated localization patterns (*DYNC1H1* - foci, *KIF1C* - cellular extension, *ACTN1* - random and polarized) were well separated.

Interestingly, the t-SNE plot also captures the heterogeneity we had observed during visual inspection of the data. First, for genes with non-random localization, not all cells have a distinct mRNA localization. For instance, most *CEP192* cells form a clearly separated cluster with a strong intra-nuclear localization (Fig. 3a, region 2), while other *CEP192* cells are random and grouped together with genes also showing a random localization pattern (Fig. 3a, region 1). Similar observations can be made for all other genes annotated as having non-random RNA localization. Interestingly, the intra-nuclear localization pattern of *CEP192* transcripts was not among the patterns used for simulation. This shows that the features we have designed are informative beyond the simulated patterns and in principle capable of adequately describing other patterns as well. Second, some spatial RNA distributions appear to be pattern mixtures. For instance, *DYNC1H1* mRNAs form foci[11], which are displayed in the upper part of the t-SNE plot (Fig. 3a, region 3). However, in some cells these foci are located toward the nuclear envelope, and these cells are positioned in close proximity with other cells displaying nuclear localization in the t-SNE plot (Fig. 3a, region 4). We conclude that our features are capable of capturing the complex structure in the data, and that t-SNE visualization allows to explore the heterogeneous localization patterns displayed by different cell populations.

**Supervised classification of RNA localization**. Next, we investigated whether we can infer different clusters of subcellular localization at the single-cell level. k-means with 4 classes correctly separates the manually annotated localization classes (random, cellular extension, foci, and nuclear-associated) but failed to isolate a small cluster of cells with strong intra-nuclear localization (Fig. 3b). In contrast, spectral clustering was able to find this cluster but fused larger localization patterns (Fig. 3b, Supplementary Note 5). Thus, unsupervised learning with fixed number of clusters can be used in order to identify localization patterns. These results further suggest that it is important to explore the data structure in detail and in particular the large

clusters. This then allows to tailor the level of detail in the analysis to the biological question.

Next, we applied hierarchical clustering (Supplementary Note 5). This approach arranges the data based on similarity and does not require a pre-definition or inference of the number of classes. Importantly, this allows to visually explore the substructure of large clusters. It also permits to inspect which features are similar among sub-groups, providing a basis for a more informed bio-physical interpretation of clusters and sub-clusters. Hierarchical clustering revealed diverse and subtle localization patterns. For instance, we observed highly specific sub-clusters of nuclear-associated patterns corresponding to intra-nuclear localization (Fig. 3c). We also observed distinct pattern mixtures for RNA foci, with localization either towards the cell membrane or the nuclear envelope (Fig. 3d).

In summary, we show that the developed workflow is capable of describing the different manually annotated localization patterns in real experiments. We also illustrate how different non-supervised clustering methods can be used to explore the data.

**Supervised classification of experimental data**. We next addressed the question of whether the simulation framework can be used in a supervised setting. Unlike unsupervised methods, supervised learning allows to impose prior knowledge in the form of a training set, i.e., to give the algorithm the opportunity to learn which feature combinations are relevant for a biologically meaningful distinction between patterns. We found that due to the realistic nature of our simulations (Supplementary Note 5), we could train a classifier (here: Random Forests[26] on simulated data, and successfully detected the correct localization patterns in experimental data (Fig. 4a, Supplementary Note 5). As a further validation, we compared the performance of this classifier and a classifier trained on manually annotated data and found nearly identical performance (Supplementary Note 5). These results show that the feature distributions of simulated and real data are close enough to allow to train a classifier with simulated data without a notable loss in performance.

**Heterogeneity in mRNA localization**. Interestingly, with this analysis we can study heterogeneity in RNA localization in detail. As described before, such heterogeneity can correspond to the co-existence of several cellular subpopulations with pure patterns, or a mixture of patterns within individual cells. To distinguish between these two scenarios, we turned to the posterior probabilities for single cells, i.e., the probabilities of a cell to belong to each of the patterns (Fig. 4b, c, Supplementary Note 5). From these posterior probabilities we can calculate the Gini impurity at the single-cell level, which gives an indication of the purity of the pattern for that particular cell (low values indicating a very pure pattern). The Gini impurity can also be calculated on the population level, i.e., on the average posterior probabilities, indicating the heterogeneity at the population level. By plotting the Gini impurity for the population against the intracellular Gini impurity (Fig. 4d), we can further investigate the nature of the heterogeneity. This reveals that some genes are characterized by rather low heterogeneity both at the cellular and the population level, such as *RAB13* and *KIF1C*, the two mRNAs accumulating in cell extensions. For other genes, we observed low heterogeneity at the cellular level and high heterogeneity at the population level. This is the case for *CEP192*, and it indicates the existence of pure subpopulations and suggests the co-existence of different localization states (Fig. 4c). Other genes are characterized by a high intra-cellular heterogeneity, i.e., a mixture of patterns, such as *DYNC1H1* (Fig. 4c). Both population and intra-cellular

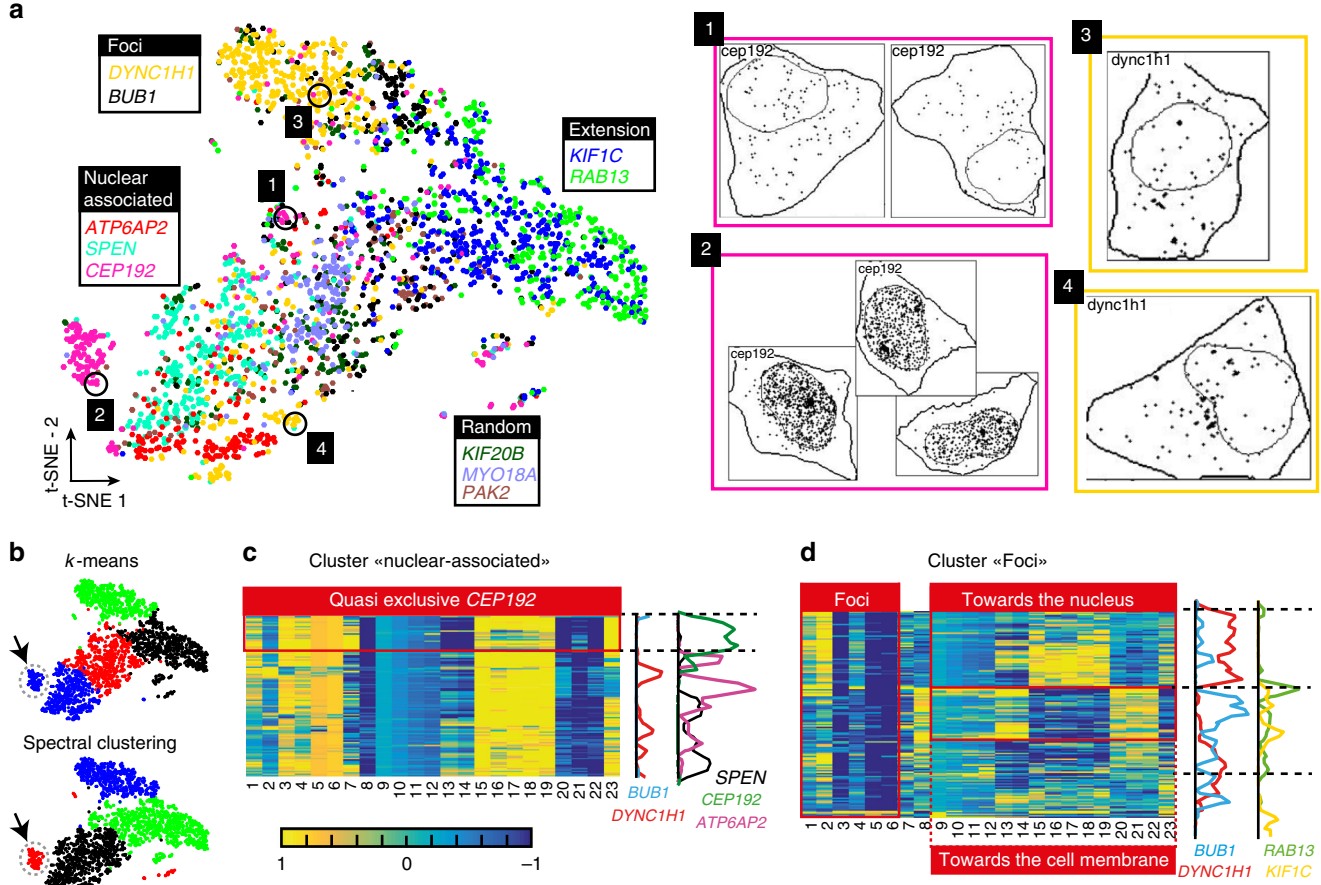

**Fig. 3** Analysis of experimental data with unsupervised approaches. **a** t-SNE projection of the 23 localization features for smFISH experiments against 10 different genes. Each dot is one cell and is color-coded according to the gene. Images on the right are examples of *DYNC1H1* and *CEP192* cells with different localization patterns from the numbered regions in the t-SNE indicated with circles. **b** t-SNE plot as in **a**, but color-coded according to results of a k-means classification or spectral clustering. The latter finds a small cluster with strong intranuclear localization (lower plot, black arrow). **c, d** Examples of hierarchical clustering results. Each row is a cell, each column a localization feature (see list of features in Table 1). Plots on the right show the smoothed distribution of the most enriched genes in the clusters

heterogeneity could imply a high degree of plasticity in mRNA localization mechanisms, possibly because of the need to rapidly adapt to changes in cell shape and cellular micro-environment. This also shows that cell-to-cell variations in gene expression occur not only on the expression level as often noted[23], but also extends to the spatial dimension.

**mRNA localization in different experimental conditions**. We finally tested how our workflow could be used to investigate changes in RNA localization. Such an analysis can allow to infer the biological relevance of an observed pattern, for instance by performing perturbation experiments. In a recent study, we reported that *DYNC1H1* displays RNA foci[11]. We further showed that these foci do not overlap with P-bodies or stress granules, but are translation factories. To test the role of translation of the formation of foci, we treated cells with Puromycin, a terminator of translation elongation. Visual examination indicates that the foci disappeared after a brief puromycin treatment (30 min). When analyzing these data with our GMM RNA detection approach, we could successfully quantify this change in localization pattern, and also show a change in the localization classes with both unsupervised and supervised approaches (Supplementary Note 5).

## Discussion

In this study, we present a framework for the simulation of smFISH images with non-random mRNA localization patterns and methods for their computational analysis. The realistic simulations allow to: (i) benchmark computational workflows aiming at studying RNA localization; (ii) assess the descriptive power of features; and (iii) evaluate the overall performance of clustering and classification methods. We developed a set of 23 localization features that could group cells based on seven simulated localization patterns with unprecedented accuracy. We further analyzed experimental data and correctly identified manually annotated localization classes. Importantly, we also demonstrate that simulated smFISH data can be used to train classifiers in a supervised setting, which when applied on real smFISH data allow to observe and quantify heterogeneity in RNA localization.

Manual annotation of smFISH images is complicated, time-consuming and subjective. When manually annotating RNA localization patterns, one typically would like to label cells with respect to the rule according to which RNAs are placed in the cell (e.g., localization close to a membrane or polarized localization). While visually solving this problem can be challenging, it is straightforward to generate images using this rule by simulation. With this, we do not only overcome the shortcomings of manual

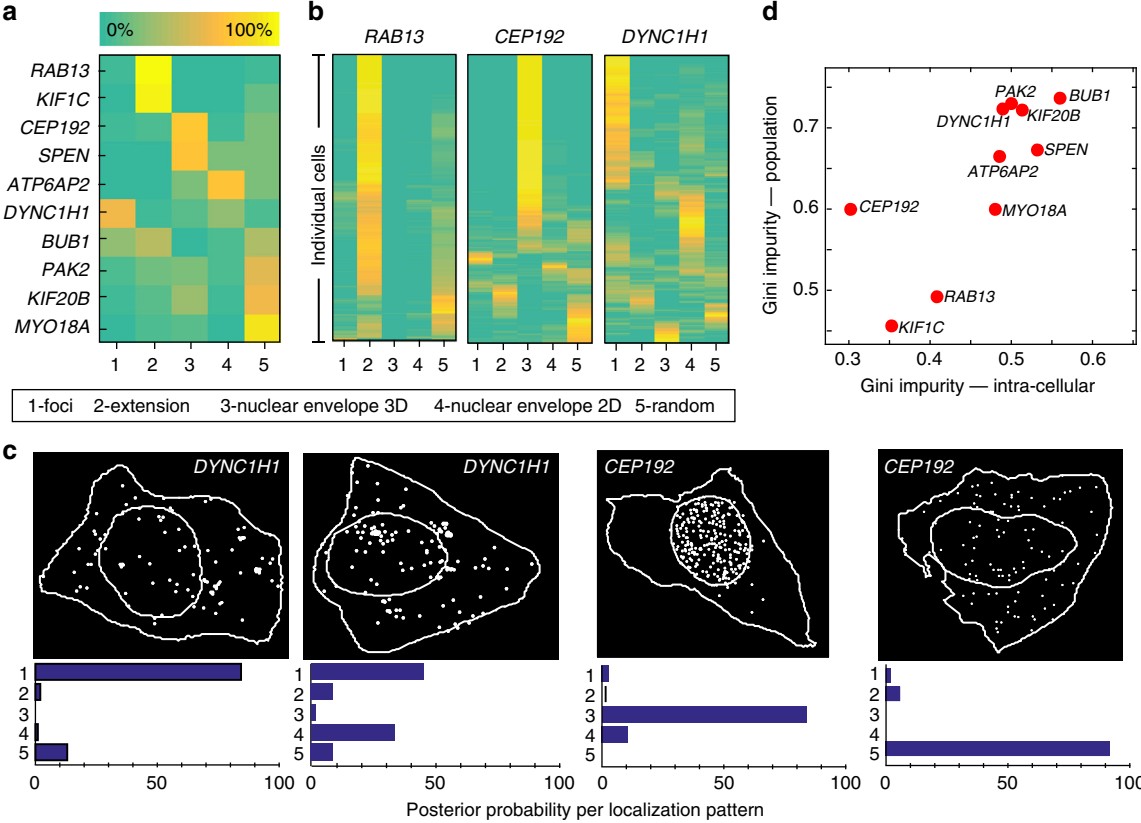

**Fig. 4** Supervised analysis of experimental data reveals localization heterogeneity. **a** Results of supervised random forest classification trained on 5 classes (mRNA foci, nuclear envelope in 2D or 3D, cell extension and random) from simulated data. Heatmap shows the majority voting results for all cells of each gene. Among the simulation classes, "Nuclear envelope 3D" is closest to the experimental class intranuclear. **b** Posterior probabilities for single cells of the indicated genes. **c** Example of individual DYNC1H1 cells and their posterior probability. **d** Scatter plot of Gini impurity calculated on average probability of a gene (population) against the average Gini impurity of individual cells of a gene (intra-cellular)

annotation, but we can also use these simulations to identify limitations of the methods and to evaluate the robustness with respect to confounders, such as the density of transcripts or the shape of cells. While we cannot exclude differences between synthetic and real images, we demonstrate that synthetic and real images are similar enough to be used for benchmarking and training.

We explore several different ways of analyzing simulated and experimental data and illustrate how the richness of RNA localization can be explored and quantified by different established statistical approaches. First, we used a qualitative exploration with t-SNE. This analysis not only provides important insights about predominant RNA localization patterns but is also informative about the subtleties in these patterns. For instance, it can show the existence of small sub-groups of cells with a very specific localization pattern, or mixtures of different patterns. Second, we used different unsupervised classification approaches (k-means, spectral clustering, and hierarchical clustering). k-means provides a good global view over the predominant localization classes but is not well suited to detect small sub-populations, even if they have a strong signature in their features. Here, spectral-clustering provides better sensitivity. Complementary to this, hierarchical clustering allows performing a detailed inspection of the data. In this analysis, the number of classes is not pre-defined or inferred and the structure of the data can be investigated to discover sub-clusters. An additional benefit is the interpretability of the features, allowing attribution of biophysical properties to the identified clusters. Lastly, we applied supervised classification with

Random Forests. We used the high quality of the simulated data to train a classifier, which could then be applied to experimental data. Supervised classification allows imposing prior biological knowledge and an advantage is to find rare localization classes that can be missed by unsupervised approaches. We also propose a way to quantitatively analyze the localization heterogeneity at the population and intra-cellular level. Which approach is used for a particular data set depends on the underlying biological question and the required level of detail necessary to answer this question.

Automatic analysis of an experimental data set consisting of 10 genes revealed different levels of heterogeneity in terms of RNA localization: heterogeneity at the population and intra-cellular level have different origins and have to be interpreted accordingly. The existence of different localization patterns in different cellular subpopulations for a given gene could be indicative of different biological states, and the localization pattern could be the approximation of these cellular states. Interpretation of pattern mixtures within a single cell is more complicated. If the patterns are mutually exclusive (for instance, if a fraction of RNA localizes at the nuclear envelope and another subset of RNA at the cell membrane), this may point to a double function of the encoded protein or a very high localization dynamic. If the patterns are not mutually exclusive (for instance organization into foci, where foci can be at different locations in the cell), this rather indicates the existence of subclasses, i.e., subtleties inside the defined classes, that have not been defined as separate classes. These subtleties could potentially bare important information, as

they might be indicative of different biological functions, e.g., by interaction with different cellular components. An important case corresponds to mRNA accumulating in foci. This generic localization class represents one of the most frequently observed patterns in *Drosophila*[3], but occurs in other model systems as well. Such foci are likely implicated in different biological processes, such as storage in P-bodies[27], vesicles[28], or specialized translation factories[11]. Their function can then be investigated by perturbation experiments, additional markers, or secondary criteria such as sub-cellular position.

While we validated and applied the workflow mainly in HeLa cells, we also successfully identified RNA localization patterns in a different cell line (C2C12 cells, Supplementary Note 5). The presented methodology should thus be applicable to other model systems. Recent studies show wide-spread RNA localization in organisms such as *Drosophila* embryos[3]. While patterns occurring at the scale of the entire embryo (or tissue) would require an extension of the proposed workflow, namely the addition of other landmarks and probably other localization features, there are number of intra-cellular localization patterns where our methodology could essentially complement these studies.

In this work, we integrated two generic cellular landmarks, namely the cytoplasm and the nucleus, which we believe are important for any large-scale localization screen. Such a screen could then reveal interesting localization patterns that are investigated in targeted follow-up experiments employing other markers tailored to the hypothesis inferred from the large-scale screen. For instance, RNA foci have been observed for a number of genes[3], but their functional role still needs to be elucidated. Such an analysis requires perturbation experiments and/or the inclusion of additional makers (Supplementary Note 5). We showed previously that *DYNC1H1* foci are neither P-bodies nor stress granules, but act as specialized translation factories. The open design of our approach allows integration of additional markers to discriminate cellular compartments.

In summary, we provide a framework for simulation of synthetic smFISH images and validated methods and tools for the analysis of intracellular RNA localization. We validated this workflow on simulated data and demonstrated how different statistical approaches can be used to investigate the complexity and richness of smFISH images and therefore provide methods and tools to explore the spatial dimension of gene expression inside cells.

## Methods

**Cell culture**. HeLa and C2C12 cell lines were obtained from ATCC and grown in DMEM medium (Gibco) supplemented with 10% FCS (Sigma). HeLa cell lines stably transfected with GFP-tagged BAC[29] were a kind gift from Anthony Hyman laboratory (MPI-CBG, Dresden, Germany) and were grown in DMEM medium supplemented with 10% FCS and 0.4 mg/ml G418 (Gibco).

**smFISH probes**. smiFISH probes were purchased from Integrated DNA Technologies. smFISH probes were synthesized by J.M. Escudier (SPCMIB, Toulouse, France) and labeled with Cy3 mono-reactive dye pack (GE Healthcare). Probe sequences are available in Supplementary Data 1.

**smFISH experiments for cell shape**. Single molecule FISH (smFISH) and our recently published inexpensive variant (smiFISH[13]) were performed as follows. Briefly, cells were fixed with 4% paraformaldehyde (Electron Microscopy Sciences) for 20 min at RT, and permeabilised in 70% ethanol overnight at 4 °C.

smiFISH against GAPDH, *RAB13, KIF1C, DYNC1H1, ACTN1* was performed in HeLa or C2C12 cells against the endogenous mRNAs, using two types of probes: (i) 24 unlabeled primary probes containing both a mRNA targeting sequence and a shared sequence (FLAP); (ii) a secondary probe conjugated to two Cy5 moieties (for *GAPDH*) or two Cy3 moieties (for *RAB13, KIF1C, DYNC1H1, ACTN1*), pre-hybridized in vitro at 65 °C to the primary probes via the FLAP sequence. For all other mRNAs, HeLa cell lines stably expressing a GFP tagged version of the gene of interest, which was expressed from a bacterial artificial chromosome (BAC)[29], were used. These BACs carry the entire gene regulatory sequences and in particular the

mRNA untranslated regions often required for proper mRNA localization. smFISH was performed against the GFP-IRES-Neo sequence of the BAC, with a pool of 40 oligonucleotide probes, carrying up to four Cy3 fluorophores each. For *BUB1, RAB13, KIF1C*, a CellMask™ channel was additionally recorded. Mock-smFISH was performed in HeLa cells with the above cited pool of probes, targeting an artificial sequence (GFP-IRES-Neo), not expressed in the cells.

All FISH experiments were performed overnight at 37 °C in a buffer containing 1X SSC, 15% formamide (Sigma), 10% dextran sulfate (Sigma), 0.34 mg/ml tRNA (Sigma), 0.2 mg/ml BSA (Roche Diagnostics), 2 mM VRC (Sigma). The next day, the samples were washed at 37 °C in 1× SSC, 15% formamide. In the specified cases, the cells were labeled with HCS CellMask™ Green Stain (Molecular Probes) diluted to 50 ng/ml in PBS, for 5 min following FISH. Cells were subsequently washed with PBS, and mounted with Vectashield mounting medium with DAPI (Vector Laboratories).

**Imaging to infer cell shape**. Three-dimensional image stacks were captured on a wide-field microscope (Zeiss Axioimager Z1) equipped with a 63 × 1.4 NA objective and a scMOS camera (Zyla 4.2 MP, Andor Technology) and controlled with Metamorph (Version 7.8.8.0; Molecular Devices).

**Image analysis to infer cell shape**. 2D segmentation of nuclei and cells was performed with the DAPI and CellMask™ channels with the open-source software CellCognition[19] using a standard segmentation workflow: Otsu and watershed separation for nuclei in the DAPI channel. Each nucleus then serves as a seed for a watershed segmentation to obtain the cells in the CellMask™ channel. Individual GAPDH mRNA molecules were localized with FISH-quant[14] in 3D. The Matlab function boundary was used to determine the conforming 3D boundary around the mRNAs, corresponding to the 3D cellular outline. GAPDH is predominantly excluded from the nuclei. To infer the positioning and height of the nuclei, the z-positions of mRNAs localized in small region in the center of the nuclei were analyzed. mRNAs were automatically separated into two groups above and below the nucleus by k-means clustering. This allowed inference of the average 3D height and position of the nuclei. For more information, see Supplementary Note 1.

**mRNA detection and localization in simulations**. mRNA detection in 2D was performed with the MATLAB script provided in Battich et al.[7].

mRNA detection in 3D was performed with a standard spot detection approach in FISH-quant[14]. In short, an approximation of the second derivative of the images was first calculated with a LoG filter. mRNAs were pre-detected with a local maximum detection with a manually determined intensity threshold. After detection, spots are fitted with a 3D Gaussian function and the GMM approach applied as detailed in Supplementary Note 2. In short, the algorithm reconstructs the mRNA foci in multiple single mRNA by using the average signal of a single mRNA as a reference.

For each cell, the different localization features are calculated in Matlab, and saved as a text file that can be further analyzed. Ripley-K function calculation was implemented from scratch. The Matlab functions pdist and p_poly_dist (from Matlab file exchange) were used to compute the distance between mRNAs and a reference point (cell centroid, nucleus centroid) and mRNA and a polygon (cell membrane, nuclear envelope). The Matlab function imopen was used to perform a morphological opening on the cellular outlines. The normalization coefficients were calculated using the bwdist function that compute the distance map of a binary mask.

**Statistical methods to analyze localization features**. t-SNE projections were performed using the tsne function implemented in MATLAB available at https://lvdmaaten.github.io/tsne/. k-means classification was performed using the Matlab function kmeans using 50 replicates; spectral clustering with the function SpectralClustering from Matlab file exchange; hierarchical clustering with the Matlab function clustergram with linkage method ward. Random forest (RF) classification was performed with the Matlab function treebagger. 100 trees were trained on simulated cells with moderate and high pattern strength and mRNA densities corresponding to an average of 100 and 200 mRNAs/cell. Gini impurity was calculated from the posterior probabilities for the RF classification: either from the mean probability per mRNA (population impurity) or the mean impurity for all cells of a mRNA (intracellular impurity).

**Cell segmentation for mRNA localization**. Automated nuclear and cell segmentation was performed with a custom algorithm based on the U-net[30] deep convolutional network. Nuclear segmentation was performed with the DAPI channel, cell segmentation was performed either with CellMask™ (if present) or with the actual smFISH image. For segmentation, 3D images were projected into 2D images with a recently described approach[13].

**mRNA detection and feature calculation**. Analysis of experimental smFISH data was performed as described for simulated data. Except that the detection intensity threshold was determined for each experiment (mRNA) separately. We implemented a simple Matlab user interface FQ_detect, which is now part of FISH-

quant, to facilitate the determination of this threshold. It proposes an automatically calculated threshold based on Otsu thresholding, based on empirically determined criteria. We refer to the FISH-quant documentation for more details.

**Code availability**. We provide the entire Matlab code for the analysis described in this paper, which we integrated into our smFISH analysis package FISH-quant[14] available at https://bitbucket.org/muellerflorian/fish_quant

## Data availability

Data used to simulate images are available on Zenodo https://doi.org/10.5281/zenodo.1413488. Additional experimental data supporting the findings of this study are available from the corresponding authors upon reasonable request.

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

## Acknowledgements

We thank Anthony Hyman for providing the HeLa cell lines transfected with BAC. We thank Maria Vera for critical reading of the manuscript. This research was supported by the Agence Nationale de la Recherche (ANR-11-BSV8-018-02 and ANR-14-CE10-0018-01), Institut Pasteur, Fondation pour la Recherche Médicale (FRM DBI20131228556), and La Ligue Nationale Contre le Cancer.

## Author contributions

F.M., T.W., and E.B. conceived the project. F.M. and T.W. supervised Au.S. Au.S. and F.M. developed the Matlab package to simulate and analyze RNA localization data. R.C., Ad.S., AM.T., M.P. and F.M. performed the smFISH experiments and analyzed the localization patterns with E.B. Au.S. F.M. and T.W. analyzed the data. C.Z. provided suggestions on data analysis. W.O. implemented the cell segmentation with neural networks. F.M. and T.W. wrote the manuscript.
