## [Peer Review File · Nature Communications]

Reviewers' comments:

Reviewer #1 (Remarks to the Author):

In their manuscript, Samacoits and colleagues describe an analysis framework of smFISH data that allows creation of realistic simulated images for training machine learning algorithms that in return are capable of reliably classifying RNA localization patterns observed in real smFISH images, as demonstrated by the authors. By introducing novel descriptors and careful normalization, they manage to cluster 6-7 out of the 8 localization patterns with high accuracy and have acceptable results for the remaining one-two categories. When applying the learning algorithms trained by the simulated images, correct localization pattern is assigned for 9 out of the 10 test transcripts analysed in cultured cells. This relatively low number of test transcripts unfortunately does not allow the reader to judge whether there is a uniform performance of the trained classifiers for all the different localization patterns. As suggested by the analysis of the simulated data, some patterns may be more difficult to recognize than others, thus I'd recommend to increase the number of test transcripts to about 6-9 per localization pattern.

Besides this the presented work is methodologically solid and appears to have a great potential. However, this is what I find the greatest shortcoming of the manuscript: it lacks any discussions - or equivalent parts - that would introduce the reader this potential. As the number of observed - not to mention possible - RNA localization patterns exceed greatly that analysed in this manuscript if one considers biologically/functionally relevant reference systems (e.g. apicobasal polarity, organelles etc.), several questions come to mind: how would the introduction of such biological references - through observable markers - change the performance of the classification? How many different patterns could be reliably distinguished? How different two patterns should be to be distinctly classified?

This latter is of particular interest, as typically perturbation screens - e.g. drugs or other chemical/ physical conditions - are performed in cultured cells. How small changes could be reliably detected? Could the changes in the cell-to-cell heterogeneity of the observed localization pattern be used as a measure of effect (possibly)?

How simple/difficult would be to implement a similar framework for a different model system, such as for tissues, organs or organisms (e.g. embryos) where much more RNA localization patterns are possible to be observed (see e.g. <http://fly-fish.cabr.utoronto.ca> or <http://tomancak-srv1.mpi-cbg.de/DOT/main>)?

In my view, discussing and possibly demonstrating the usefulness of the described framework, this manuscript would gain a much broader interest in the field of RNA biology. This discussion should be based on data analysis, where possible, e.g. :

1. a more upfront analysis of the effects of the pattern strength used as a measure of a scale of change/pattern difference that can be recognized
2. Testing the effects of a biological reference, e.g. the apical domain and/or mitochondria that were shown/implicated to be actively involved in RNA localization

Minor points:

In Figure 3b lower panel "a small cluster with strong intranuclear localization (lower plot, black arrow)" should be visible, however I could not find the mentioned arrow (nor in the corresponding Figure S5-3d).

In Note 5 (page 36 of the supplement) the authors mention 12 genes, but later they talk about only the listed 10 transcripts. I suspect this was a typo.

sincerely

Imre Gaspar
EMBL Heidelberg

Reviewer #2 (Remarks to the Author):

A quantitative framework to study sub-cellular mRNA localization
Samacoits et al.

Subcellular mRNA localization is an important mechanism used by cells to temporally and spatially regulate gene expression. In this paper, Samacoits et al. claim to have developed a comprehensive tool for sub-cellular mRNA localization assessment in fixed cells. This work relies on the solid smFISH technology that is currently used for quantitative gene expression analysis in many laboratories, making this tool of wide and timely interest. They simulate smFISH images to build a data-set for mRNA localization classification by supervised or unsupervised learning. Finally, they used their model on experimental data and conclude that mRNA localization in an immortalized cell line has high level of heterogeneity by single cell analysis.

Specifically, the mRNA localization analysis can be subdivided into three major steps: 1) Simulation of smFISH data that takes into consideration variables such as cell shape, mRNA concentrations and mRNA localization patterns 2) Analysis of the simulated data using 2D t-SNE projections to automatically distinguish mRNA localization patterns 3) Application of the analysis tool to experimental data generated for 10 different genes.

This tool would potentially be of value for the RNA localization field. However, the submitted manuscript and program require major improvements as to utility for a broad audience. At this stage the program suffers from technical issues that render some of the conclusions questionable and the manuscript is difficult to read and is in need of a rewrite. For instance, the main text lacks a clear outline, making the flow hard to follow. The tutorial associated with the program is an extension of the supplemental material and does not help the user to understand how to actually use the program. Due to the lack of sufficient guidance, we couldn't test this program in the time allotted for the review process.

To improve the clarity of the manuscript we suggest the following changes:

Major points:

- The manuscript lacks a proper discussion about the biological relevance and applications to other cell types. For example, the authors could compare a wt to a mutant version of a localized mRNA. The biologicals results should be put in perspective: which genes have which patterns, what does that represent, can it be regulated (cell type, induction, etc)?
- To put their paper in larger context, the authors should also compare their method to other published mRNA localization tools (eg. Park HY et al. Cell Rep. 2012; Lécuyer E. Cell 2007, but there are many others).
- The biological conclusion generated with this work is that there is high degree of heterogeneity for mRNA localization at the single cell level. This result can be interpreted in different ways: it could be due to real biological heterogeneity, to variability in smFISH efficiency or to an intrinsic limitation of the analysis framework, such as the definition of the localization patterns. The authors should provide more examples of smFISH data, not only one cell at the time. For instance, they could show 3D pictures for multiple cells for a single mRNA to help reader to infer the origin of the heterogeneity.
- The simulation step is an essential part of this tool and it is based on several assumptions. One point that should be further discussed is the impact of the mRNA concentration. The authors arbitrarily decide to simulate mRNAs ranging from 50 to 400. For this reason, it is unclear whether the model would work also for genes that are not expressed in that range such as GAPDH, HSP70 or RPB1.
- Another point that should be discussed is the impact of the cellular shape on the localization efficiency. The authors analyze how differences in cell geometry for their specific cell type (Hela cells) affect mRNA localization. However, a validation with a completely different cell type, such as neurons or macrophages, could extend the use of this tool to different cell shapes.
- It is known in the literature that immortalized cell lines do not localize mRNA, or do so poorly, so this tool is developed on a cell of little relevance for the field. The "heterogeneity" may simply be variations in the noise that they have selected statistically to be presented as significant. The

extensive development of the algorithms using a cell line consequently ignores the biological questions for which it is supposedly designed. They should use their analysis on primary cells.

- Another important aspect that is not clearly explained in the text is the definition of the mRNA localization patterns. Many parameters are decided arbitrarily such as the strength of the phenotype. It is unclear how these arbitrary parameters could impact on the quality of the model and how reproducible these definitions would be from lab to lab.
- The information about how to use the software is too fragmented. To make this tool useful for a wide range of users, the authors could provide a clear outline of a typical workflow, starting with the experiments required to generate the data and a step-by-step protocol for the localization analysis.
- In an attempt to use the program, we faced the following issues: the program ran for a very long time (about 12 hours) and it crashed because of a memory issue. It would be useful to have clear specifications about the system requirements. Furthermore, the program doesn't clearly provide information about the progression status (such as cell numbers analyzed). We are sophisticated users of image analysis routines, and have developed some of our own, so if we have trouble with this tool, imagine the problem for the general user.

Minor points

- The authors developed a new and potentially useful tool to quantify mRNA accumulated in foci. To validate the quantifications, we suggest the use of two-color smFISH with two sets of probes recognizing the same mRNA (i.e. DYNC1H1). Probes in each of the two color sets should provide similar quantifications.
- The authors assumed that mRNA density is only proportional to the cell volume, ignoring other parameters such as stochasticity of gene expression or effects of the microenvironment. These parameters should be also considered, especially for the localization of inducible mRNAs.

Reviewer #3 (Remarks to the Author):

The manuscript titled "A quantitative framework to study sub-cellular mRNA localization" by Samacoits et al. describes an integrated analysis framework of smFISH data for studying sub-cellular mRNA localization.

The manuscript is accompanied by quite extended supplementary material (about 50 pages) where all aspects of their approach are described in great detail. The overall approach appears to be scientifically sound. However, the rationale behind this strategy is not clear: The authors explain that the main difficulty would be the "absence of a ground truth for smFISH" and therefore argue to start with a simulation framework to "generate realistic ground truth data mimicking experimentally observed mRNA localization". The authors need to explain and justify in much more detail, why it should not be possible to generate ground truth from existing microscopy images. To give just one example, the authors are referred to the recent work by Long et al. performing quantitative mRNA imaging throughout the entire *Drosophila* brain (<https://www.nature.com/articles/nmeth.4309>).

Why can such type of data not serve as a ground truth for the validation of an analysis framework described here?

Is there a particular reason why this or other type of 3D image data are not manually analyzed to check the accuracy of the computational framework?

As it is presented now, it appears to me that the authors simply did not have access to or did not team up with experimentalists to design their analysis framework. In fact, the fairly complex analysis framework could be much more streamlined by being based on ground truth data. In its current state, I wonder whether it will be ever useful to the scientific community; apart from the fact that most of the code seems to be written in the commercial software environment MATLAB

that is not widely accessible in experimental labs. My impression is that the usefulness of this computational tool will be highly limited due to its complexity and design.

If indeed it would not be possible to build and validate the analysis framework on ground truth data, then it does not seem appropriate to refer to the synthetically generated data in this study as being "realistic". It is at least not clear, what is meant by such a statement: How would unrealistic synthetic data look like? The authors need to explain their approach in a more rigorous way making it (sound?) less illogical.

Even though the supplementary material is very extended and describes the approach in great detail, I missed a table where all parameters of the analysis are summarized. As a consequence, it is not clear to me how many parameters there are.

Moreover, the parameters that are used all need justification and where possible even a comment with regard to their sensitivity regarding the analysis itself. For example, it is not sufficient to just state "An mRNA is considered to be localized at the nuclear membrane if it is below a fixed distance (800 nm) from the membrane..." (page 6 of supplementary material and this is repeated later on, e.g. page 7). Unfortunately, no justification why the parameter value of 800 nm should be reasonable is given.

In conclusion, the complexity of the computational framework that is largely based on commercial software is overwhelming making its usefulness highly questionable, all the more that it remains unclear how realistic the synthetic cell representation really is. This main issue could only be solved if real data would be manually analyzed and used as a ground truth.

Minor issues:

The authors should carefully check and correct their manuscript, including the supplementary material, for various typos and mistakes. For example:

- 1) Supplement, page 15, Fig S2-4 c: Label in the plot: not -> no
- 2) Supplement, page 19, last sentence of Section 2.5: There is one "and" too much in this sentence.
- 3) Supplement, page 21, top of page: The averaging formula should be presented as an equation (do not just give the right hand side).

General reply

We thank the reviewers for their comments. It is our pleasure to submit an extensively revised manuscript, which addresses all raised concerns and questions. The improvements fall into the following three categories:

- 1) We **provide new experimental data and simulations** to further strengthen previous validations, and illustrate the general applicability of the approach, e.g. experiments on more genes, different cell lines, sensitivity analysis and impact of drug treatments.
- 2) We **provide an improved source code**. First, we improved the code to speed-up calculations and to provide more direct feed-back about the current status of the analysis. Second, we provide a didactic step-by-step tutorial integrating the various suggestions of the reviewers.
- 3) We **substantially expanded the main text**. The original manuscript was formatted as a *Brief Communication*. This short format did not allow us to properly explain important aspects of our method, which then raised numerous questions of the reviewers. In the current manuscript, we now better position our method with respect to existing literature and available data, discuss its possible applications and also describe its limitations.

Below we provide detailed answers to all comments in blue italic text.

Reviewer #1	1
Reviewer #2	8
Reviewer #3	16
References	21

Reviewer #1

In their manuscript, Samacoits and colleagues describe an analysis framework of smFISH data that allows creation of realistic simulated images for training machine learning algorithms that in return are capable of reliably classifying RNA localization patterns observed in real smFISH images, as demonstrated by the authors. By introducing novel descriptors and careful normalization, they manage to cluster 6-7 out of the 8 localization patterns with high accuracy and have acceptable results for the remaining one-two categories. When applying the learning algorithms trained by the simulated images, correct localization pattern is assigned for 9 out of the 10 test transcripts analysed in cultured cells.

RL.1 This relatively low number of test transcripts unfortunately does not allow the reader to judge whether there is a uniform performance of the trained classifiers for all the different localization patterns. As suggested by the analysis of the simulated data, some patterns may be more difficult to recognize than others, thus I'd recommend to increase the number of test transcripts to about 6-9 per localization pattern. Besides this the presented work is methodologically solid and appears to have a great potential.

First, we would like to thank the reviewer for this overall positive assessment of our methodological work. We fully agree with the reviewer that our simulations suggest that we should not expect constant performance across different localization classes. We mention this in the main paper (page 5). We would like to emphasize however, that these classification difficulties concern mainly the cellular level, and much less so the gene level. Given the heterogeneity in RNA localization for any given gene, it would be very challenging (and probably simply wrong) to assign a gene to one single localization pattern. In the manuscript, we propose instead to assign to each gene a vector with the probabilities to belong to each localization pattern. This attenuates the differences in classification accuracy, as there is no hard assignment at the gene level. At the single cell level however, it

seems unavoidable to have different accuracies for different classes (which is also the case for computational phenotyping).

We therefore envision that the actual classification task will be performed at the cell level, and that localization at the gene level will be defined by the probabilities to belong to each localization class. We provided in the original analysis experimental data for 150-400 cells per localization pattern (2600 cells total) which is in line with studies for classification of protein localization patterns. In order to further demonstrate the performance of the method, we have now also compared the results to manual annotation of localization patterns at the cellular level and achieved a pooled accuracy of 83% (see comment R3.7 below). We therefore conclude that our method is capable of assigning localization classes at the cellular level with high accuracy and can therefore be used to assign to each transcript a vector of localization probabilities.

Nevertheless, we agree with the reviewer that similar localization patterns should be classified similarly, irrespective of the gene identity. In other words, we need to check that cells from different genes but with similar localization patterns cover overlapping regions in the feature space. In the submitted manuscript, we have provided pairs of genes with overall similar localization patterns, and we have shown that they indeed cover overlapping regions in the feature space. It should be noted that one difficulty in analyzing more genes is that few localized mRNAs have been described in cell lines besides mRNAs localizing on the ER, mitochondria, or in cell extensions (Battich et al., 2013; Chen et al., 2015; Mili et al., 2008). The reason is not that regular cell lines do not contain localized mRNAs, but that most of the studies so far focused on embryos or highly polarized cells such as neurons or epithelial cells. Indeed, we currently perform a screen in HeLa cells where we analyzed more than 500 mRNA by smFISH, and found 32 with specific localization patterns, most being unrelated to ER, mitochondria and cell extensions (unpublished data).

[REDACTED]

[REDACTED]

However, this is what I find the greatest shortcoming of the manuscript: it lacks **any discussions** -or equivalent parts - that would introduce the reader this potential. As the number of observed – not to mention possible - RNA localization patterns exceed greatly that analysed in this manuscript if one considers biologically/functionally relevant reference systems (e.g. apicobasal polarity, organelles etc.), several questions come to mind:

R 1.2 how would the introduction of such biological references – through observable markers – change the performance of the classification?

Introducing additional marker could enhance the performance and sensitivity of the approach. For instance, we know that the class "RNA foci" groups several subclasses: some of these foci correspond to P-bodies (as in the case of CEP170P1; unpublished data), or to other cellular structures (as in the case of DYNC1H1; see Pichon et al., 2016). Inclusion of a P-body marker may thus be helpful to define RNA localization more precisely. However, we believe that specific markers show their full potential in the context of targeted secondary screens aiming at a detailed study of genes with particular localization patterns (for instance after their identification in a primary large-scale screen). For instance, adding a P-body marker would for instance not help to define the "nuclear-associated" subclasses. Our workflow is generic in that it can accommodate newly designed features into the simulation and learning framework we propose, including additional markers (see below). Adding specific marker would typically require the development of tailored analysis tools to fully exploit the additional

marker image. These could be added to the workflow in order to derive new features (such as distances to other cellular landmarks, e.g. P-bodies, or a query protein). In this case, classification performance is very likely to increase. Technically, this implies (1) segmentation, e.g. detection of the position of all P-bodies, (2) definition of positional features with respect to the segmentation result, e.g. distance to the detected P-bodies. Depending on the nature of the additional channel, this can be straightforward or relatively complicated. Technically, we already do this for two generic markers (cell membrane by CellMaskTM, and nuclear envelope by DAPI), which we and others assume to be important in the general case for any large-scale screening application. Nevertheless, we feel that adding more specific markers would be beyond the scope of this paper, as we do not aim at testing a specific biological hypothesis (e.g. RNA localization in P-bodies). We now explain this in more detail in the manuscript (main text page 8).

R 1.3 How many different patterns could be reliably distinguished? How different two patterns should be to be distinctly classified? This latter is of particular interest, as typically perturbation screens – e.g. drugs or other chemical/ physical conditions - are performed in cultured cells. How small changes could be reliably detected? Could the changes in the cell-to-cell heterogeneity of the observed localization pattern be used as a measure of effect (possibly)?

These are important but also complex questions, the answers to which have now been integrated in the manuscript and the supplementary notes (main text page 5, Supplementary Note 4.7 and 5.5).

*In general, there is no clear answer to the question “**How many different patterns are in the data?**”. There is no generally accepted measurement for difference between image patterns, as there are many ways in which patterns could potentially differ. Importantly, depending on the biological question, it is often useful to give preference to a certain aspect of localization differences. In our workflow, we have several ways in which we can influence the sensitivity of our method: (1) the features used [if a very particular aspect is to be investigated, one could add some dedicated feature] (2) the machine learning approach including parameters that can be set according to the user’s interest (in case of unsupervised learning, for instance the number of classes) (3) the training set (in case of supervised learning), where the important feature differences are inferred from the annotation (or in our case from the simulations). We can therefore not answer this question in general, but we provide different explanations and analyses on the potential capacity of our approach.*

***Defining the number of classes** is a reoccurring and general problem in machine learning. There is no theoretical limit for the number of classes. This being said, it is also true that increasing the number of classes for the same dataset generally leads to poorer performance, as more subtle decisions must be made and there are more chances for misclassifications. We feel that the final choice of “how many classes?” should remain at the discretion of the researcher. While there are methods to automatically determine the number of classes from a purely algorithmic point of view (Supplementary Note 4.6), we feel that it is a bad idea to blindly trust such methods, simply because from a biological point of view, this question cannot be answered unequivocally. As an example, in our data both BUB1 and DYNC1H1 transcripts form mRNA foci, which can be detected as a localization pattern with our method. Additionally, other features describing distance distributions to the cell membrane and the nuclear envelope, reveal that these foci in turn display different sub-cellular localization themselves. Depending on the scope of the study, we could either consider these genes to be in the same class (foci) or in separate classes (foci at the cell periphery and foci at the nuclear envelope). Different machine learning approaches can be used to explore and quantitatively describe these observations.*

*We show that **hierarchical clustering** (which has no fixed number of classes) reveals the existence of large localization classes with a stable signature in the localization features. These classes can be subdivided in order to reveal subtler sub-patterns (Fig R3 shows the mentioned example of foci). This illustrates how hierarchical clustering can be used to inspect the data in order to define the localization classes based on the biological questions. Further, it also shows how to inspect the links between features and clustering allow for a more informed bio-physical decision if a further distinction in more classes is desirable or not.*

*We also illustrate how **supervised classification** with soft assignment, i.e. a cell is assigned to not only one localization pattern, can provide insights into potential sub-classes of localization patterns. In these approaches (here we used Random Forest, but we could have used any state-of-the-art method), a classifier is trained on known localization patterns. When we applied such a classifier to experimental data, we found that some cells are best described with a mixture of patterns (we refer to this as intra-cellular localization heterogeneity, Fig R3, right). This then indicates more fine-grade distinctions in localization patterns than was covered by the initial class definition.*

Figure R3. (Left) Hierarchical clustering of experimental data from four mRNAs. Shown is only cluster corresponding to mRNA foci (features 1-6). Features 13-22 describe different sub-cellular localization. (Right) Random forest classification of experimental data (from Main text). Shown are two examples for *DYNC1H1*. The graphs below show the posterior probabilities, which indicate how well each cell is described by each of the 5 trained localization classes. (1-foci, 2-extension, 3-nuclear envelope 3D, 4-nuclear envelope 2D, 5-random).

In summary, while there is no theoretical limitation to the number of classes and no unequivocal method to infer this number from the data, we show how different machine learning approaches provide a detailed view that can guide the researcher to infer the most suitable number of classes for the biological question at hand. We now explain the different strategies in much more detail in the Discussion section of the main text (page 8).

Detecting subtle differences in localization patterns

We now include a new analysis illustrating to which degree we can still distinguish each pattern. For a detailed description of these results, we refer to the Supplementary Note 4.7. In short, we used our simulation framework to analyze the impact of the pattern strength on the outcome of the analysis. In this note, we show how to use the simulation framework for sensitivity analysis, providing an intuition on the differences between patterns that can still be distinguished (Fig R4). Such an application could prove particularly useful in the case of perturbation studies, where we want to analyze the effect of a drug or a gene silencing experiment, and we therefore need to understand how well subtle pattern differences can be distinguished. We have added a corresponding paragraph to the main text (page 5).

Figure R4. t-SNE projection of simulated cells of random and 3 non-random localization classes (foci, localization towards the cell membrane in 2D (cell2D), and localization towards the nuclear envelope in 2D (nuc2D)). The color shade corresponds to pattern strength. For random localization, there is no pattern strength parameter defined. Circles correspond to 70, 90, 95 and 99 percentiles of random localization.

R 1.4 How simple/difficult would be to implement a similar framework for a different model system, such as for tissues, organs or organisms (e.g. embryos) where much more RNA localization patterns are possible to be observed (see e.g. <http://fly-fish.ccb.utoronto.ca> or <http://tomancak-srv1.mpi-cbg.de/DOT/main>)?

We completely agree that mRNA localization in other biological system is a very interesting application of our workflow, provided that cell can be segmented and nuclei are stained. The Krause and Tomancak lab beautifully demonstrated that mRNA localization in *Drosophila* is very common (Jambor et al., 2015; Lécuyer et al., 2007; Wilk et al., 2016). The authors showed a wide-range of localization patterns, some of which we believe could be readily analyzed with our proposed method. While patterns occurring at the scale of the entire embryo (or tissue) would require an extension of the proposed workflow, namely the addition of other landmarks and probably other localization features, there are number of intra-cellular localization patterns where our methodology could essentially complement existing studies. For instance, Wilk et al. reports that “One of the

most common localization categories observed during later developmental stages is “cytoplasmic foci”, which we specifically consider in our analysis. The authors further note that the “The numbers of foci per cell and their subcellular locations also varied significantly between different transcripts”. In our data, we already showed for *DYNC1H1* and *BUB1* how different sub-cellular localization of foci could be identified (Fig R3, left). For other sub-cellular localizations observed by Wilke et al, e.g. towards the membrane facing parasegmental grooves for instance, adequate additional markers would be needed. Frequently detected were also different “perinuclear” localizations, e.g. with either foci or a polarized localization. Our workflow provides several features describing the distance of mRNA to the nucleus and also the degree of polarization. We hence believe that many of these localization patterns could be accurately quantified and/or classified. Other than that, our simulation workflow could be used to design and validate new features. Importantly, these studies also describe localization heterogeneity by having shared annotations, e.g. foci with different sub-cellular localizations. Our framework now provides the possibility for a quantitative description of this heterogeneity. We now extended the manuscript to clarify these points (main text page 8-9).

In my view, discussing and possibly demonstrating the usefulness of the described framework, this manuscript would gain a much broader interest in the field of RNA biology. This discussion should be based on data analysis, where possible, e.g. :

R 1.5 a more upfront analysis of the effects of the pattern strength used as a measure of a scale of change/pattern difference that can be recognized

We would like to refer to the comment R1.3 above, where we considered the suggestion of the reviewer.

R 1.6 Testing the effects of a biological reference, e.g. the apical domain and/or mitochondria that were shown/implicated to be actively involved in RNA localization

This question is related to R1.2 and partly to R1.4. Additional adequate markers can certainly provide important insights, yet they are usually tailored to more specific biological questions and follow-up experiments, which are beyond the scope of this article. Conceptually, the integration is reasonably straight-forward, but technically this requires adding developments specific for each marker (new segmentation, new features, ...), as explained above.

Minor points:

R 1.7 In Figure 3b lower panel “a small cluster with strong intranuclear localization (lower plot, black arrow)” should be visible, however I could not find the mentioned arrow (nor in the corresponding Figure S5-3d).

We added the arrows to these figures. We would also like to point out that we provide a zoom-able version of this t-SNE plot (Fig 3a), where we visualize the detected RNA spots on top of the cell and nucleus segmentation. The individual cells are spatially arranged to match the location of their t-SNE projection:

https://muellerflorian.github.io/locFISH_deepzoom/#results/tsne_exp

We now emphasize this plot in the manuscript, as well as the tools to generate such plots (main text page 5).

R 1.8 In Note 5 (page 36 of the supplement) the authors mention 12 genes, but later they talk about only the listed 10 transcripts. I suspect this was a typo.

We corrected this typo.

Reviewer #2

Subcellular mRNA localization is an important mechanism used by cells to temporally and spatially regulate gene expression. In this paper, Samacoits et al. claim to have developed a comprehensive tool for sub-cellular mRNA localization assessment in fixed cells. This work relies on the solid smFISH technology that is currently used for quantitative gene expression analysis in many laboratories, making this tool of wide and timely interest. They simulate smFISH images to build a data-set for mRNA localization classification by supervised or unsupervised learning. Finally, they used their model on experimental data and conclude that mRNA localization in an immortalized cell line has high level of heterogeneity by single cell analysis.

Specifically, the mRNA localization analysis can be subdivided into three major steps: 1) Simulation of smFISH data that takes into consideration variables such as cell shape, mRNA concentrations and mRNA localization patterns 2) Analysis of the simulated data using 2D t-SNE projections to automatically distinguish mRNA localization patterns 3) Application of the analysis tool to experimental data generated for 10 different genes.

This tool would potentially be of value for the RNA localization field. However, the submitted manuscript and program require major improvements as to utility for a broad audience. At this stage the program suffers from technical issues that render some of the conclusions questionable and the manuscript is difficult to read and is in need of a rewrite. For instance, the main text lacks a clear outline, making the flow hard to follow. The tutorial associated with the program is an extension of the supplemental material and does not help the user to understand how to actually use the program. Due to the lack of sufficient guidance, we couldn't test this program in the time allotted for the review process.

To improve the clarity of the manuscript we suggest the following changes:

Major points:

R 2.1 The manuscript lacks a proper discussion about the biological relevance and applications to other cell types. For example, the authors could compare a wt to a mutant version of a localized mRNA. The biologicals results should be put in perspective: which genes have which patterns, what does that represent, can it be regulated (cell type, induction, etc)?

We now address a number of these concerns in the revised manuscript.

- 1. We provide a more detailed description of biological relevance of the mRNA localization for different biological systems in the introduction (main text page 1).*
- 2. We provide additional data where we successfully applied our approach to another cell type (see comment R2.5 below). In the manuscript, we now discuss in more detail to what extent our approach can be applied to other cell types, or other biological model systems such as drosophila (see comment R1.4 above).*
- 3. We provide an analysis from data from a perturbation experiment (see Fig R below), and how our workflow can provide a quantitative read-out in the change of a localization pattern (see comment R1.3 above).*
- 4. We discuss some of the data in light of biological questions, for instance how Puromycin treatment supports the conclusion that the observed pattern of DYNCIH1 mRNA foci are translation factories (see below and in main paper page 7 and 9). We would like to recall however that the **main focus of this study is to provide an analysis framework to investigate mRNA localization**. The biological data serves as validation to show that this approach can be used for experimental data, and further how the observed heterogeneity can be quantitatively analyzed. We develop this aspect in the paper because this is both new and highly relevant for the biology of RNA localization. Indeed, it suggests that this process is much more plastic and dynamic than previously suspected, as least in cell lines, and this likely relates to the micro-environment of individual cells. This is a nice addition to the growing body of evidences indicating high cell-to-cell variability in gene expression, and it goes beyond the traditional quantitative aspects of RNA expression levels and extends these concepts to the spatial dimension.*

Analysis of mRNA localization in different experimental conditions

We did not analyze a WT vs. a mutant mRNA, but we performed new experiments to show that mRNA localization for DYNCIH1 is translation-dependent. In a recent study, we reported that DYNCIH1 mRNA concentrates in foci (Pichon et al., 2016). We further showed that these foci do not overlap with P-bodies or stress granules, but that they are translation factories (Pichon et al., 2016). To test the role of translation in the formation of foci, we treated cells with Puromycin, a terminator of translation elongation. Visual examination

indicates that the foci disappear after a brief treatment (30 minutes) with Puromycin (Fig R5a). Such a change can be quantified directly, by comparing the most impacted localization features. For this purpose, we calculated the average number of foci per cell, and how many mRNAs are per foci (Fig R5b). We can also analyze these data with the unsupervised and supervised analysis methods described in the paper. Fig R5c shows a t-SNE plot, where untreated and Puromycin-treated cells occupy different regions of the plot. As an alternative, we also analyzed these data with our supervised classification approach (Fig R5d). In agreement with the analysis above, we observe a loss of localization in foci after Puromycin treatment. In summary, this illustrates how the established framework can be used to study perturbation experiments, and it demonstrate that the formation of *DYNC1H1* foci is translation-dependent. These data challenge an established paradigm, which states that RNA localization is translation-independent. We show these data now in the Supplement (Note 5.6) and discuss these results in the paper 7 and 9.

Figure R5. (a) Representative images of *DYNC1H1*: untreated (left) and after Puromycin treatment (right). Shown are maximum intensity projections, and the segmented cell and nuclei. Same scaling was applied to all images to allow direct comparison. (b) Boxplots show comparison of foci in *DYNC1H1* mRNA localization before and after Puromycin treatment: average number of foci per cell, and average number of mRNAs per foci. (c) t-SNE analysis of *DYNC1H1* smFISH data. Each dot is one cell. Color-code is according to molecular identity: untreated (red) and after treatment with Puromycin (blue). (d) Supervised classification of *DYNC1H1* smFISH data. Left histogram shows probability distribution among different localization patterns for untreated cells, right histogram for cells after treatment with Puromycin.

R 2.2 To put their paper in larger context, the authors should also compare their method to other published mRNA localization tools (eg. Park HY et al. Cell Rep. 2012;., but there are many others).

We agree that the original manuscript did not review all existing approaches and tools in the field, which was due to the strict length limitations imposed by the original submission format (Brief Communication). In the revised manuscript, we now describe existing studies and tools (main text page 1, 2, 3, 4). We also clearly separate the question of studying intra-cellular mRNA localization vs. localization at the scale of a tissue/embryo (main text page 1; see also comment R1.4 above). To our knowledge, the most advanced analysis for intra-cellular mRNA localization was performed by the Pelkmans lab (Battich et al., 2013; Stoeger et al., 2015). We used this approach as the initial reference for our benchmarking. Park et al. focuses on a very particular localization pattern (polarization). The authors propose two localization features, which are also part of our proposed feature set. Due to space limitations, this was detailed in the supplement in the previous version of the manuscript. We have moved this now to the main text (main text page. 4).

R 2.3 The biological conclusion generated with this work is that there is high degree of heterogeneity for mRNA localization at the single cell level. This result can be interpreted in different ways: it could be due to real biological heterogeneity, to variability in smFISH efficiency or to an intrinsic limitation of the analysis framework, such as the definition of the localization patterns. The authors should provide more examples of smFISH data, not only one cell at the time. For instance, they could show 3D pictures for multiple cells for a single mRNA to help reader to infer the origin of the heterogeneity.

We agree that such a manual inspection is very important, and something we also use routinely in a first assessment of the data. For this purpose, we also provide the possibility to create a zoom-able t-SNE plot, where the data-points are replaced by 2D thumbnail images of cellular and nuclear outlines, and the detected mRNAs. This representation allows for convenient navigation. We agree with the reviewer that the visualization of smFISH data and the corresponding detection results is a challenging and important aspect and we hope that this kind of plots will help researchers in analyzing their data. We discuss this now in the main text (main text page 5,6). For an example of such a plot, please use the following link:

https://muellerflorian.github.io/locFISH_deepzoom/#results/tsne_exp

In the revised manuscript, we provide a dedicated figure for the analysis of the observed heterogeneity with more examples (Main Fig 4). We also added a more detailed discussion of the potential causes and implications of heterogeneity (main text page 8,9).

R 2.4 The simulation step is an essential part of this tool and it is based on several assumptions. One point that should be further discussed is the impact of the mRNA concentration. The authors arbitrarily decide to simulate mRNAs ranging from 50 to 400. For this reason, it is unclear whether the model would work also for genes that are not expressed in that range such as GAPDH, HSP70 or RPB1.

RNA expression levels vary substantially between different genes and even between cells for a given gene. To test the robustness of our approach towards these variations, we simulated a range of expression levels. This range was not arbitrarily chosen, but covers the observed expression levels from a recent large-scale smFISH study (Battich et al., 2013; Fig R6 below). We now reference this in the main text (page 3).

Fig R6. (Left) Histogram of simulated mRNA levels after pooling all 4 expression regimes (Right) Histogram of measured mRNA levels from >900 genes with smFISH and RNA-seq (from Battich et al., 2013).

However, as the reviewer points out, some genes show substantially higher expression levels, such as GAPDH. These highly expressed genes are challenging for smFISH in general, since individual mRNAs become difficult to spatially resolve because of the high local densities. Such high densities may also impact some of the localization features. In particular, we speculate that the Ripley's L-function, which measures spatial clustering and dispersion, could be impacted. To more rigorously test this, we now performed simulations with highly expressed genes (800 on average per cell, but with some cells having up to 1500-2000 mRNAs (see Fig R7 below). We pooled these simulations with a second set with an average of 200 mRNAs per cell, in order to assess classification of high and low expressing cells (Fig R7). The confusion matrix for this pooled data-set reveals that correct clustering was achieved for almost all localization patterns, despite the wide range of expression levels. However, and as speculated, foci were commonly misclassified as random. Closer inspection of these results revealed that: (i) the mis-classified cells were those with high expression levels, (ii) they were mixed with random since the characteristic "foci-signature" of the Ripley-features disappeared. These simulations show that the expression level indeed impacts the detection accuracy of some localization patterns, but also that most patterns are not affected. We explain these limitations in the manuscript (main text page 5, and Supplementary Fig S4-5).

Figure R7. a Histogram of mRNA levels after pooling simulations with an average of 200 and 800 mRNAs per cell. **b** Confusion matrix of analysis of the pooled cell population.

R 2.5 Another point that should be discussed is the impact of the cellular shape on the localization efficiency. The authors analyze how differences in cell geometry for their specific cell type (Hela cells) affect mRNA localization. However, a validation with a completely different cell type, such as neurons or macrophages, could extend the use of this tool to different cell shapes.

Indeed, our method aims at identifying localization patterns. We therefore see cell morphology as a confounder, even if analyzing the link between morphology and RNA localization is an interesting question in itself. As suggested by the reviewer, we further tested the performance of our workflow on a different cell type. For this comparison, we choose C2C12 cells, which are an immortalized mouse myoblast cell line. HeLa cells are human epithelial cells derived from an adenocarcinoma, and have a different morphology and size: C2C12 are larger and less round compared to HeLa cells (Fig R8).

Figure R8. Comparison of size and shape of HeLa and C2C12 cells. Parameters were estimated from 2D segmentation masks. The “shape factors” gives an indication of the objects shape, it is calculated as $4 \cdot \pi \cdot \text{area} / (\text{perimeter}^2)$. A perfect circle has a value of 1, while a thin thread-like object has the lowest shape factor approaching 0.

We performed smFISH against 3 genes with different localization patterns: *Dync1h1* (foci), *Kif1C* (cell extension), *Actn1* (random). We analyzed the data with the exact same analysis workflow as used for HeLa cells. The t-SNE analysis (Fig R9) shows that as expected, the cells are projected to different regions in the t-SNE space, corresponding to their specific localization pattern. We intended ACTN as a control, as we had manually annotated that localization was random. After our quantitative analysis, we observed that there was indeed a large subpopulation with random localization (Fig R9), but there was also another subpopulation showing polarized localization, which has then been validated by manual inspection (Fig R9). These results demonstrate that the established workflow is well suited to distinguish localization patterns in other cell lines than HeLa. We refer to these results in the main text (page 6) and added the detailed analysis to the Supplement (Note 5.2).

Figure R9. *t*-SNE plot of experimental data for C2C12 cells. Each data-point is a cell colored according to the gene identity of the visualized mRNA. Images are maximum intensity projections with cell outlines shown in red, and nuclear outlines shown in yellow.

R 2.6 It is known in the literature that immortalized cell lines do not localize mRNA, or do so poorly, so this tool is developed on a cell of little relevance for the field. The "heterogeneity" may simply be variations in the noise that they have selected statistically to be presented as significant. The extensive development of the algorithms using a cell line cosequently ignores the biological questions for which it is supposedly designed. They should use their analysis on primary cells.

Although primary cells may display more and/or stronger localization patterns, we respectfully disagree with the statement that immortalized cell lines do not localize mRNAs. Immortalized cell lines have been successfully used to study mRNA localization. RNA localizing in cellular extensions have been discovered in NIH/3T3 (Mili et al., 2008; and follow-up papers), HeLa and IMR-90 cells have been used in a large smFISH screen (Battich et al., 2013; Cabili et al., 2015; Chen et al., 2015), which describe mRNA localizing in the perinuclear region, in a polarized manner, or near the cell edges. We now include these references in the introduction (page. 1). We would also like to refer to one of our recent papers (Pichon et al., 2016), where we initially observed mRNA foci for DYNC1H1 in HeLa cells, and then confirmed this localization in all of the rodent and human cells that we analyzed: NIH3T3, U2OS, HEK, and primary neurons. The fact that these foci are translation factories and dissolve upon translation inhibition demonstrate that they are functionally relevant. While these foci are not P-bodies, it should be noted that RNA localization in P-bodies is quite common (2000 mRNAs were found in P-bodies in HEK293 cells (Hubstenberger et al., 2017).

[REDACTED]

It is also important to note that our mRNA detection method and our features are not cell-type specific, and that the workflow has been applied to another cell type (see comment R2.5 above). We are thus confident that it could be also used in primary cells.

R 2.7 Another important aspect that is not clearly explained in the text is the definition of the mRNA localization patterns. Many parameters are decided arbitrarily such as the strength of the phenotype. It is unclear how these arbitrary parameters could impact on the quality of the model and how reproducible these definitions would be from lab to lab.

We now expanded in the manuscript to explain how the patterns are simulated and what role they play in the development of the analysis workflow and also for the analysis of experimental data. For the convenience of the reviewer, we also provide a condensed explanation here.

Role of simulations

We use simulations primarily to benchmark feature descriptions and clustering methods. These features are then calculated on experimental data and used for clustering. In this setting, the impact of pattern strength and other simulation parameters is minor, as the parameters impact only the generation of the benchmark data, but not the actual classification and feature calculation. The situation is different however in the case of supervised learning, as the discriminative function is learned from the training set. We added Supplementary Note 4.7 in order to demonstrate the effect of choosing different pattern strengths for the training set. There is an effect on classification performance. In particular, if the generated examples are too extreme, i.e. much easier to classify than real samples, the learned rule will be suboptimal. We would like to stress however, that this is a general problem of supervised learning even if samples were manually annotated: if they do not represent the “difficult cases”, the learned classifier will be suboptimal. Altogether, we see that there is a good agreement between supervised and unsupervised methods indicating that our pooling strategy is providing robust results even in the case of supervised learning.

Parameters in simulations

In order to simulate patterns, we establish mathematical rules with some parameters that control how to place mRNAs in the 3D space. As pointed out above and in comment R3.4 below, while the goal is to simulate “realistic” images (i.e. visually similar images with similar feature distributions), we do not claim to provide an accurate mathematical or biophysical description of the experimental data. In other words, we accept that the distribution of simulated data will not exactly match the distribution of real data.

Each pattern is described by two types of parameters. First, some general properties that remain fixed for a given pattern, such as the distance to the nuclear envelope. Given a reasonable choice of these parameters, we expect little impact on the classification performance. For instance, if we train a classifier to distinguish patterns where many RNA have a distance of less than 800 nm from the envelope, we are confident that the same classifier would work for patterns where many RNAs have a distance of less than 1200 nm: the overall feature distribution is not likely to change dramatically (see also comment R3.6 below). The second group is what we called “pattern strength”. These parameters control how “strong” a pattern will be, e.g. which fraction of RNAs display the pattern. With this parameter, one can control how difficult the classification task will be for the simulated data. We chose to simulate a range of pattern strengths and to report results for the individual values separately and for the pooled data.

Altogether, there is no impact of simulation parameters on the method in the unsupervised setting other than the reported accuracy (but not the actual accuracy), and there is some impact in the supervised setting. In this case, we need to check that (1) simulated and real images are visually similar and (2) real and simulated cells tend to localize in similar regions in the feature space.

R 2.8 The information about how to use the software is too fragmented. To make this tool useful for a wide range of users, the authors could provide a clear outline of a typical workflow, starting with the experiments required to generate the data and a step-by step protocol for the localization analysis.

We now provide such a more detailed documentation, including flow-charts with input/output of the different steps and the different parameters. For the revision, our Matlab package (locFISH) is available from the Dropbox link listed below. This folder also contains the PDF of the updated user manual.

<https://www.dropbox.com/sh/fz0vyath35dao4c/AADNDk5TPE-vtJ7mKlOeP3fia?dl=0>

Please note that upon publication the software will be distributed via bitbucket. This will allow efficient handling of issues and distribution of further improvements of the source code.

R 2.9 In an attempt to use the program, we faced the following issues: the program ran for a very long time (about 12 hours) and it crashed because of a memory issue. It would be useful to have clear specifications about the system requirements. Furthermore, the program doesn't clearly provide information about the progression status (such as cell numbers analyzed). We are sophisticated users of image analysis routines, and have developed some of our own, so if we have trouble with this tool, imagine the problem for the general user.

We apologize for the problems encountered by the reviewer. We improved several aspects of the code to facilitate its use:

- *We updated the helpfile and provide more information about the workflow and tutorials for how to get started (R2.8).*
- *We specify the configuration of the computer used to generate these data in the documentation.*
- *We improved the code to simulate smFISH images resulting in a speed-up of 10-100 fold depending on the localization pattern.*
- *A small library of cell shapes is now directly distributed with the software package. This allows to test the simulation workflow immediately without the need to download the larger library with all cell shapes.*
- *There are two time-consuming steps in the workflow (1) simulating smFISH images and (2) spot detection and feature calculation. For either step, we now provide a status bar indicating progress and also an autosave feature. The latter allows to continue processing if the analysis crashed.*

Minor points

R 2.10 The authors developed a new and potentially useful tool to quantify mRNA accumulated in foci. To validate the quantifications, we suggest the use of two-color smFISH with two sets of probes recognizing the same mRNA (i.e. DYNC1H1). Probes in each of the two color sets should provide similar quantifications.

We performed these experiments as suggested. We synthesized a total of 60 probes against DYNC1H1, split them into two pools with different fluorophores (25 probes with CY3, 35 probes with CY5), and recorded the different channels (Fig R10A). We analyzed each channel separately with the automated workflow described in the Supplement. We then compared the estimated number of mRNAs in each focus for the two channels (Fig R10B). The estimates are in very good agreement (Pearson's correlation coefficient $R = 0.74$), confirming the reliability and accuracy of our approach. We also included these results in the supplement of the revised manuscript (Supplementary Fig S2-8).

Figure R10. A) A focal plane of a HeLa cell with DYNC1H1 mRNA being targeted with two probe-sets labeled with CY5 and CY3. Right image shows overlay and DAPI stain. B) The quantification results for detected RNA foci ($N=169$). Red line is diagonal with slope 1 indicating perfect correlation.

R 2.11 The authors assumed that mRNA density is only proportional to the cell volume, ignoring other parameters such as stochasticity of gene expression or affect of the microenvironment. These parameters should be also considered, especially for the localization of inducible mRNAs.

We completely agree that large fluctuations in RNA density occur in real biological data for the various reasons stated by the reviewer. Our initial text was an over-simplification and it has now been corrected (see main text

page 3). Indeed, ongoing research efforts by many labs aim to understand the origin and implications of these variations, and they include the identification of both intrinsic and extrinsic factors. Please note that when studying mRNA localization, we have to guarantee that the workflow is **independent** of the mRNA copy number. However, for the purpose of validating the workflow and identifying RNA localization patterns, it is not the origin of these variations that is important, but the fact that the classification is not sensitive to these fluctuations.

To consider this in our workflow, we simulated different expression regimes and analyzed the pooled data (see also R2.4). Our results illustrate that even in the pooled data, we can identify cells with similar localization patterns even if they differ in their expression level. We now extended the main text, to describe the motivation of this analysis and its implications in more detail (using also the examples put forward by the reviewer). We now explain this in more detail in the revised text (main text page 3, 4, 5).

Reviewer #3

The manuscript titled "A quantitative framework to study sub-cellular mRNA localization" by Samacoits et al. describes an integrated analysis framework of smFISH data for studying sub-cellular mRNA localization.

R 3.1 The manuscript is accompanied by quite extended supplementary material (about 50 pages) where all aspects of their approach are described in great detail. The overall approach appears to be scientifically sound. However, the rationale behind this strategy is not clear: The authors explain that the main difficulty would be the "absence of a ground truth for smFISH" and therefore argue to start with a simulation framework to "generate realistic ground truth data mimicking experimentally observed mRNA localization". The authors need to explain and justify in much more detail, why it should not be possible to generate ground truth from existing microscopy images.

We would like to thank the reviewer for his/her appreciation of the technical quality of the paper. While the reviewer qualifies the approach to be "scientifically sound", (s)he is concerned about the usefulness of the simulation approach. The reviewer assumes that it is fairly easy to provide high-quality manual annotations for smFISH data. This is understandable, because the mainstream approach for morphological phenotyping is clearly learning from annotated data (Held et al., 2010; Jones et al., 2009; Neumann et al., 2010). On a side note we would like to point out that one of the lead authors was very active in developing this kind of approaches in the past (T. Walter, see Held et al., 2010; Neumann et al., 2010). Learning from annotated ground truth data was therefore clearly the first thing we also had in mind when we started the project.

We learned however that this strategy is difficult in the case of RNA localization and even more so for large scale applications. In many cases, it turns out that localization patterns are much more difficult to interpret than morphologies. There are many reasons for that:

- *The human visual system is good at evaluating an overall morphology, but less good at integrating the positions of point clouds in varying reference volumes and comparing this to random localization. Indeed, if you look long enough at a randomly distributed set of points, you often tend to see a pattern. Another problem is that the assessment of localization should be independent from the number of points (expression level). This is very hard to achieve for a human. Another complicating aspect is that localization patterns can mix at the single-cell level. All of this makes the manual annotation very time consuming, difficult and highly subjective.*
- *The 3D nature of the problem makes it complicated and time consuming to navigate through the data sets. Indeed, some localization patterns are difficult to appreciate by looking at standard projections. Of course, one could load every volume in a 3D viewer and rotate the point clouds in order to discover patterns, but it would be a major time investment to annotate larger data sets in this way, and actually far beyond any realistic time commitment. There will certainly be new developments and visualization techniques allowing for a more seamless navigation and annotation, but at the moment, we simply lack the informatic tools to manually discover and annotate some of the patterns.*
- *Moreover, quantitative studies of sub-cellular RNA localization by smFISH is a new field. Whereas for cellular and nuclear morphologies, there is a huge body of prior knowledge and decades if not centuries of experience we can rely on when manually interpreting them, this is certainly not true for RNA localization patterns. This means that in many cases and in particular for large-scale screening approaches, we look for unknown patterns. This is also partly the case in large-scale morphological screening, but to a much lesser extent.*

This being said, it is indeed possible to manually annotate a subset of localization patterns in a reasonable time frame, in particular when we restrict ourselves to clear cases. But we do not think that this is the way to go in the future. Also, we are not aware of any publication performing supervised learning from manual annotations on RNA localization data.

The difficulty for our visual system is that we try to solve an inverse problem: we observe a cloud of points, and we wish to categorize the patterns according to the putative rule with which this point cloud was generated. Our approach to overcome this problem is to build a forward model that generates data from rules, and then use these data to benchmark features and pipelines, study the robustness and limitations, and analyze how the workflow deals with pattern mixtures. As we control the generation of the patterns, we have better means to study the behavior and performance of the algorithms. Given the problems of manual annotations, we do not see

any reason why we should trust manual annotations more than data that were generated according to plausible rules of RNA localization.

Even if the exact generating process is not part of the "forward model" of real data (e.g. in case of an unknown pattern that was not simulated), the features will probably still be informative enough to solve the clustering / classification problem. For this we have no guarantee, but the detection of the intra-nuclear pattern, which was not included in the simulation, suggests that this actually works (Main Fig 3a, cluster 2 of CEP192 mRNA). In any case, this potential problem also holds for supervised learning from annotated samples, as we can never be sure of having predefined all classes in a data set.

Following the reviewer's suggestion, we have added now an extensive justification for simulations to the main text (page 1,2). It is indeed crucial for the article to clearly convey this message, and we would like to thank the reviewer for pointing to this weakness of the initial manuscript.

R 3.2 To give just one example, the authors are referred to the recent work by Long et al. performing quantitative mRNA imaging throughout the entire Drosophila brain (<https://www.nature.com/articles/nmeth.4309>). Why can such type of data not serve as a ground truth for the validation of an analysis framework described here? Is there a particular reason why this or other type of 3D image data are not manually analyzed to check the accuracy of the computational framework?

The article "Quantitative mRNA imaging throughout the entire Drosophila brain" by Long et al. describes a method to perform smFISH in the Drosophila brain with subcellular resolution. While RNA localization in cells can be observed in these data, subcellular localization is clearly not the main target of the study and the experimental methods have not been optimized to address this question (for instance, more markers would be needed to calculate localization features at the cellular level). Importantly, there is no ground truth associated to this article, in terms of sub-cellular localization. This is actually a general problem: the scientific community is certainly not lacking data, but it is in need for more annotated data. Regarding the difficulty of annotation of RNA localization data, we would like to refer to our answer to R3.1 above.

On a different note, it is of course interesting to think about possible extensions of our method to make it applicable in the context of organs or organisms. Regarding this, we would like to refer to our answer R1.4.

R 3.3 As it is presented now, it appears to me that the authors simply did not have access to or did not team up with experimentalists to design their analysis framework. In fact, the fairly complex analysis framework could be much more streamlined by being based on ground truth data. In its current state, I wonder whether it will be ever useful to the scientific community; apart from the fact that most of the code seems to be written in the commercial software environment MATLAB that is not widely accessible in experimental labs. My impression is that the usefulness of this computational tool will be highly limited due to its complexity and design.

We deeply regret that our manuscript created this impression. As a matter of fact, more than half of the authors are experimental scientists that are familiar with smFISH and RNA localization for more than 15 years (see Basyuk et al., 2003; Bertrand et al., 1998; Fusco et al., 2003 by lead author E. Bertrand). The project has been designed in a collaborative effort to address the most pressing practical problem in High Content Screening for RNA localization today, and the data presented in this article has been generated by the authors.

[REDACTED]

*Regarding MATLAB: while it is correct that **Matlab** is a commercial software, we disagree with the reviewer that this makes the usefulness of our approach questionable. Matlab is widely used for image processing and readily available in many labs or imaging platforms. Many published papers by the NATURE PUBLISHING GROUP are accompanied by Matlab source code. As an example, our software package FISH-quant to detect mRNAs in smFISH, is written in Matlab, and widely used in many research labs (66 citations since its publication in 2013; source: web of science).*

Regarding the complexity of the workflow. We would like to point out that the application of our method with the proposed feature set and in an unsupervised setting is actually straightforward. We used the simulation framework to design and benchmark feature sets and the machine learning workflow, but the simulation is not

required for an unsupervised analysis. Regarding the usefulness and justification of the simulation part, we would like to refer to answer R3.1.

R 3.4 If indeed it would not be possible to build and validate the analysis framework on ground truth data, then it does not seem appropriate to refer to the synthetically generated data in this study as being "realistic". It is at least not clear, what is meant by such a statement: How would unrealistic synthetic data look like? The authors need to explain their approach in a more rigorous way making it (sound?) less illogical.

First of all, as mentioned in R3.1, we can manually annotate images (see R3.7 below), but it does not seem to be a scalable approach in this context.

A pragmatic, yet subjective definition of realistic images would be that in a side-by-side comparison, it is not immediately obvious which images are experimental and which are simulated. This was tested on experienced microscopists (e.g. FM, EB and MP), and it can be directly appreciated from the example images that we provide. In addition, we compared experimental and simulated data in terms of the extracted localization features. For many patterns we observe very similar feature distributions (See Supplementary Fig S5-9), indicating that the images are also realistic with respect to the extracted localization features.

R 3.5 Even though the supplementary material is very extended and describes the approach in great detail, I missed a table where all parameters of the analysis are summarized. As a consequence, it is not clear to me how many parameters there are.

We now extended the helpfile describing the use of the program (see R2.8 above). We provide a visual overview of the different steps and the main required parameters in the user manual. We do not believe that a big table with all parameters will be of much use, however. The parameters concern different steps of the workflow and do not need to be jointly optimized or tweaked. For instance, some parameters are used for cell segmentation and RNA detection, other parameters determine how different features are calculated. Regarding simulation parameters, we would like to refer to R3.6.

R 3.6 Moreover, the parameters that are used all need justification and where possible even a comment with regard to their sensitivity regarding the analysis itself. For example, it is not sufficient to just state "An mRNA is considered to be localized at the nuclear membrane if it is below a fixed distance (800 nm) form the membrane..." (page 6 of supplementary material and this is repeated later on, e.g. page 7). Unfortunately, no justification why the parameter value of 800 nm should be reasonable is given.

This question refers to parameters that are only used only for the simulations. We would like to emphasize again that the parameters used in the simulation have no impact on the features calculated on real data (see also comment R2.7). In the unsupervised setting (which is most likely the predominant application case), there is therefore no impact on the performance of the algorithm. If really extreme values were chosen for the simulation (in this example 0 nm or 20 µm), they might be useless for benchmarking, because the resulting images would either be too easy to classify (e.g. a distance of 0 nm from the membrane with standard deviation of 0) or too different from the actual pattern they are supposed to represent (e.g. if the RNA were at a distance of 30% of the cell width, they could hardly be considered as localizing close to the membrane). If the parameters of the simulation are set in a reasonable range, there should not impact on the actual algorithm, only on the reported accuracy.

Second, the situation is different in the supervised setting, where we learn from the simulated data. Here, our guess was that the impact of most parameters would be minor. In the example mentioned above, our guess was that the classifier will still learn the relevant signatures, whether we choose 800nm or 1200nm, as the feature profile will vary little for these values as compared to feature vectors for different localization patterns.

It is important to stress that we do not intend to provide a perfect biophysical model for RNA localization that takes into account the biophysical processes inside the cell. The simulation rather plays the role of data augmentation in the machine learning literature, where we provide reasonable examples for the different classes. In other words: the question is not whether the true distance between the membrane and the RNAs localizing at the membrane is 800 nm or 1200 nm, but to find a method that distinguishes the class "close to the membrane" from the other patterns, such as "random localization" or "foci".

Nevertheless, to illustrate this point further and to try to convince the reviewer, we simulated images with localization towards the cell membrane in 2D with different values of the specified distance (800nm as before

and 1200nm). We then compared these simulations with other localization patterns (cell membrane in 3D, nuclear envelope, and random Fig R11, left). *k*-means clustering reveals that the 2 different simulations with a localization towards the cell membrane 2D are grouped together: even if we impose $k=5$ (i.e. we force the algorithm to perform an additional split), the two classes corresponding to the different parameter settings are almost undistinguishable. If we choose $k=4$ (Figure R11, right), we see that the cells for both cell membrane 2D simulations are clustered together. We therefore conclude that the impact of this parameter is minor on the type of analysis we are performing.

Figure R11. *k*-means clustering of 5 different localization patterns as indicated on the x-axis. (a) Clustering with 5 classes shows that the simulations towards the cell membrane are often clustered together. (b) Clustering with 4 classes shows that the simulations towards the cell membrane in 2D are grouped together and separated from the other three classes.

R 3.7 In conclusion, the complexity of the computational framework that is largely based on commercial software is overwhelming making its usefulness highly questionable, all the more that it remains unclear how realistic the synthetic cell representation really is. This main issue could only be solved if real data would be manually analyzed and used as a ground truth.

To address this point, we performed a comparison of simulation versus manual annotations. We trained a Random Forest classifier on simulated data for 5 classes (Foci, cell extension, nuclear envelope in 2D, nuclear envelope in 3D, and random), and we applied it to experimental data. We then performed manual annotations on the experimental data. The confusion matrix between these two annotations is shown below in Fig R12. We observe that there is generally good agreement between the two methods. The two main cases of disagreement concern “foci” which are confused with “localization at the nuclear envelope in 2D” (which is due to foci in close proximity to the nuclear edge) and “cell extension” which are sometimes confused with “random” indicating the difficulty of assessing how significant an accumulation in the extension is. However, the overall agreement is quite good and comparable to performances used in morphological phenotyping (see for instance Neumann et al., 2010). We note however, that it cannot be concluded that the algorithm has an accuracy of 83%, as the manual annotations can be questioned as much as the results obtained by automatic classification.

Fig. R12. Comparison of automated classification with random forest and manual annotation. Cells were manually annotated and automatically classified by a classifier trained on simulated data. Shown is the confusion matrix between this manual annotation and automated classification.

Minor issues:

R 3.8 The authors should carefully check and correct their manuscript, including the supplementary material, for various typos and mistakes. For example:

- 1) Supplement, page 15, Fig S2-4 c: Label in the plot: not -> no
- 2) Supplement, page 19, last sentence of Section 2.5: There is one "and" too much in this sentence.
- 3) Supplement, page 21, top of page: The averaging formula should be presented as an equation (do not just give the right hand side).

We corrected these and other typos in the manuscript.

References

- Basyuk, E., Galli, T., Mougel, M., Blanchard, J.-M., Sitbon, M., and Bertrand, E. (2003). Retroviral genomic RNAs are transported to the plasma membrane by endosomal vesicles. *Dev. Cell* 5, 161–174.
- Battich, N., Stoeger, T., and Pelkmans, L. (2013). Image-based transcriptomics in thousands of single human cells at single-molecule resolution. *Nat. Methods* 10, 1127–1133.
- Bertrand, E., Chartrand, P., Schaefer, M., Shenoy, S.M., Singer, R.H., and Long, R.M. (1998). Localization of ASH1 mRNA particles in living yeast. *Mol. Cell* 2, 437–445.
- Cabili, M.N., Dunagin, M.C., McClanahan, P.D., Biaesch, A., Padovan-Merhar, O., Regev, A., Rinn, J.L., and Raj, A. (2015). Localization and abundance analysis of human lncRNAs at single-cell and single-molecule resolution. *Genome Biol.* 16, 20.
- Chen, K.H., Boettiger, A.N., Moffitt, J.R., Wang, S., and Zhuang, X. (2015). Spatially resolved, highly multiplexed RNA profiling in single cells. *Science* aaa6090.
- Fusco, D., Accornero, N., Lavoie, B., Shenoy, S.M., Blanchard, J.-M., Singer, R.H., and Bertrand, E. (2003). Single mRNA molecules demonstrate probabilistic movement in living mammalian cells. *Curr. Biol. CB* 13, 161–167.
- Held, M., Schmitz, M.H.A., Fischer, B., Walter, T., Neumann, B., Olma, M.H., Peter, M., Ellenberg, J., and Gerlich, D.W. (2010). CellCognition: time-resolved phenotype annotation in high-throughput live cell imaging. *Nat. Methods* 7, 747–754.
- Hubstenberger, A., Courel, M., Bénard, M., Souquere, S., Ernoul-Lange, M., Chouaib, R., Yi, Z., Morlot, J.-B., Munier, A., Fradet, M., et al. (2017). P-Body Purification Reveals the Condensation of Repressed mRNA Regulons. *Mol. Cell* 68, 144-157.e5.
- Jambor, H., Surendranath, V., Kalinka, A.T., Mejsnik, P., Saalfeld, S., and Tomancak, P. (2015). Systematic imaging reveals features and changing localization of mRNAs in Drosophila development. *ELife* 4.
- Jones, T.R., Carpenter, A.E., Lamprecht, M.R., Moffat, J., Silver, S.J., Grenier, J.K., Castoreno, A.B., Eggert, U.S., Root, D.E., Golland, P., et al. (2009). Scoring diverse cellular morphologies in image-based screens with iterative feedback and machine learning. *Proc. Natl. Acad. Sci. U. S. A.* 106, 1826–1831.
- Lécuyer, E., Yoshida, H., Parthasarathy, N., Alm, C., Babak, T., Cerovina, T., Hughes, T.R., Tomancak, P., and Krause, H.M. (2007). Global analysis of mRNA localization reveals a prominent role in organizing cellular architecture and function. *Cell* 131, 174–187.
- Mili, S., Moissoglu, K., and Macara, I.G. (2008). Genome-wide screen reveals APC-associated RNAs enriched in cell protrusions. *Nature* 453, 115–119.
- Neumann, B., Walter, T., Hériché, J.-K., Bulkescher, J., Erfle, H., Conrad, C., Rogers, P., Poser, I., Held, M., Liebel, U., et al. (2010). Phenotypic profiling of the human genome by time-lapse microscopy reveals cell division genes. *Nature* 464, 721–727.
- Pichon, X., Bastide, A., Safieddine, A., Chouaib, R., Samacoits, A., Basyuk, E., Peter, M., Mueller, F., and Bertrand, E. (2016). Visualization of single endogenous polysomes reveals the dynamics of translation in live human cells. *J. Cell Biol.* 214, 769–781.
- Stoeger, T., Battich, N., Herrmann, M.D., Yakimovich, Y., and Pelkmans, L. (2015). Computer vision for image-based transcriptomics. *Methods*.
- Wilk, R., Hu, J., Blotsky, D., and Krause, H.M. (2016). Diverse and pervasive subcellular distributions for both coding and long noncoding RNAs. *Genes Dev.* 30, 594–609.

Reviewers' comments:

Reviewer #1 (Remarks to the Author):

I enjoyed reading the revised manuscript from Samacoits and colleagues. As I wrote previously, the presented work has been methodologically solid and appeared to have a great potential. With the revisions – especially with the added Discussion – this potential is immediately apparent to the reader, namely:

- the method, based on the 23 parameters describing RNA localization, can efficiently classify [Redacted] localization patterns and it can capture the heterogeneity of the localizations of transcripts (this was already clear in the previous version of the ms)
- it can be extended with other biological markers as localization landmarks
- the method can be adapted to other biological models (partly related to the point above)
- it is able to recognize changes in a localization pattern upon perturbation (e.g. drug screening). This I find a very important improvement.

- Nevertheless, this method does not – cannot – substitute but it can support the scientist in finding biologically relevant localization pattern or changes of such patterns. This is now explicitly spelled out in the manuscript.

Overall, I'm satisfied with the revised version of this manuscript from the Müller laboratory and I find that the presented method on the automated recognition of localization patterns will be of great utility not only to the RNA biology but very likely to the entire field of microscopy and image analysis.

Therefore, I recommend the manuscript from Samacoits and colleagues for publication in Nature Communications.

Only a few minor comments on editing:

Consider moving Table S5-2 into the manuscript, e.g. as part of Figure 2. I think this that as these parameters are the essence of the analysis they should be presented upfront.

Drosophila is the name of the genus and thus it should be italicized and start with a capital letter – or simply write fruitfly.

Similarly, I always found GINI impurity written as Gini impurity (as it appears in Figure 4d), as it is related -to my best knowledge – to the works of Corrado Gini.

Reviewer #2 (Remarks to the Author):

In this revised manuscript, Samacoits et al. addressed most of our concerns and made the manuscript and program much more solid. The manuscript, as well as the user manual, is much better written and pleasant to read. The authors also greatly improved the quality of the discussion.

While testing the program, we found that the simulation of the localization patterns is much faster, thanks to the improved source code. However, we noted that while trying to analyze our experimental data the program returned an error message. The error may be due to mishandling on our part but we suggest that, before publication, to explain better in the manual the type of images required. We had no other problems with the programs or the tutorial and we find that the program is now very accessible.

We think that with all these improvements, the program now represents a valuable tool for the community to use in order to analyze mRNA localization patterns, on a large scale and in different cell types.

Minor Comment:

There is a typo in Figure legend 4: the Figure legend 4 has two panels D but panel C is missing.

Reviewer #3 (Remarks to the Author):

Editorial Note: These remarks were uploaded as a PDF. The black writing corresponds to Reviewer #3's comments from the previous round of review, the blue writing corresponds to the authors' response to the previous round of review and the green writing corresponds to Reviewer #3's comments from this round of review.

Response to the Rebuttal by Reviewer #3 in Green colour

Reviewer #3

The manuscript titled "A quantitative framework to study sub-cellular mRNA localization" by Samacoits et al. describes an integrated analysis framework of smFISH data for studying sub-cellular mRNA localization.

R 3.1 The manuscript is accompanied by quite extended supplementary material (about 50 pages) where all aspects of their approach are described in great detail. The overall approach appears to be scientifically sound. However, the rationale behind this strategy is not clear: The authors explain that the main difficulty would be the "absence of a ground truth for smFISH" and therefore argue to start with a simulation framework to "generate realistic ground truth data mimicking experimentally observed mRNA localization". The authors need to explain and justify in much more detail, why it should not be possible to generate ground truth from existing microscopy images.

We would like to thank the reviewer for his/her appreciation of the technical quality of the paper. While the reviewer qualifies the approach to be "scientifically sound", (s)he is concerned about the usefulness of the simulation approach. The reviewer assumes that it is fairly easy to provide high-quality manual annotations for smFISH data. This is understandable, because the mainstream approach for morphological phenotyping is clearly learning from annotated data (Held et al., 2010; Jones et al., 2009; Neumann et al., 2010). On a side note we would like to point out that one of the lead authors was very active in developing this kind of approaches in the past (T. Walter, see Held et al., 2010; Neumann et al., 2010). Learning from annotated ground truth data was therefore clearly the first thing we also had in mind when we started the project.

We learned however that this strategy is difficult in the case of RNA localization and even more so for large scale applications. In many cases, it turns out that localization patterns are much more difficult to interpret than morphologies. There are many reasons for that:

- The human visual system is good at evaluating an overall morphology, but less good at integrating the positions of point clouds in varying reference volumes and comparing this to random localization. Indeed, if you look long enough at a randomly distributed set of points, you often tend to see a pattern. Another problem is that the assessment of localization should be independent from the number of points (expression level). This is very hard to achieve for a human. Another complicating aspect is that localization patterns can mix at the single-cell level. All of this makes the manual annotation very time consuming, difficult and highly subjective.*
- The 3D nature of the problem makes it complicated and time consuming to navigate through the data sets. Indeed, some localization patterns are difficult to appreciate by looking at standard projections. Of course, one could load every volume in a 3D viewer and rotate the point clouds in order to discover patterns, but it would be a major time investment to annotate larger data sets in this way, and actually far beyond any realistic time commitment. There will certainly be new developments and visualization techniques allowing for a more seamless navigation and annotation, but at the moment, we simply lack the informatic tools to manually discover and annotate some of the patterns.*
- Moreover, quantitative studies of sub-cellular RNA localization by smFISH is a new field. Whereas for cellular and nuclear morphologies, there is a huge body of prior knowledge and decades if not centuries of experience we can rely on when manually interpreting them, this is certainly not true for RNA localization patterns. This means that in many cases and in particular for large-scale screening approaches, we look for unknown patterns. This is also partly the case in large-scale morphological screening, but to a much lesser extent.*

This being said, it is indeed possible to manually annotate a subset of localization patterns in a reasonable time frame, in particular when we restrict ourselves to clear cases. But we do not think that this is the way to go in the future. Also, we are not aware of any publication performing supervised learning from manual annotations on RNA localization data.

I am not at all convinced by the argumentation of the authors. I agree with the authors that it is possible to manually annotate the real data; therefore, this should be done to evaluate the performance of their approach.

The fact that the authors “do not think that this is the way to go in the future”, is not a convincing argument in any way. That the authors “are not aware of any publication performing supervised learning from manual annotations on RNA localization data” is again not saying anything.

My point is that the author’s new computational approach needs a rigorous quantitative validation on real data in terms of performance measures.

The difficulty for our visual system is that we try to solve an inverse problem: we observe a cloud of points, and we wish to categorize the patterns according to the putative rule with which this point cloud was generated. Our approach to overcome this problem is to build a forward model that generates data from rules, and then use these data to benchmark features and pipelines, study the robustness and limitations, and analyze how the workflow deals with pattern mixtures. As we control the generation of the patterns, we have better means to study the behavior and performance of the algorithms. Given the problems of manual annotations, we do not see any reason why we should trust manual annotations more than data that were generated according to plausible rules of RNA localization.

I can see the point of the authors regarding errors made in the manual annotation. Nevertheless, in order to bridge the gap between artificial model data and real data, from my point of view, a quantitative validation on real data in terms of performance measures is the only way to go. Otherwise it remains completely unclear where the strength and weaknesses of their approach is, which is – from my point of view – unacceptable regarding the publication of this work.

I am quite surprised about the statement of the authors, i.e. “we do not see any reason why we should trust manual annotations more than data that were generated according to plausible rules of RNA localization”, because I think that I do not see any reason why I should trust in the practical applicability of any algorithm that was not quantitatively validated on real data in terms of performance measures.

Even if the exact generating process is not part of the "forward model" of real data (e.g. in case of an unknown pattern that was not simulated), the features will probably still be informative enough to solve the clustering / classification problem. For this we have no guarantee, but the detection of the intra-nuclear pattern, which was not included in the simulation, suggests that this actually works (Main Fig 3a, cluster 2 of CEP192 mRNA). In any case, this potential problem also holds for supervised learning from annotated samples, as we can never be sure of having predefined all classes in a data set.

The above statement is rather vague, because what we want to achieve is a certain level of accuracy that does go beyond suggestions that something may actually work. It is true that we can hardly ever be sure of anything, but that does not mean that we do not try to get as close as we can. I get the feeling that the authors either try to avoid the work associated with the standard approach of validation, or they have reasons to foresee that quantitative discrepancies can be expected.

Following the reviewer’s suggestion, we have added now an extensive justification for simulations to the main text (page 1,2). It is indeed crucial for the article to clearly convey this message, and we would like to thank the reviewer for pointing to this weakness of the initial manuscript.

R 3.2 To give just one example, the authors are referred to the recent work by Long et al. performing quantitative mRNA imaging throughout the entire Drosophila brain (<https://www.nature.com/articles/nmeth.4309>). Why can such type of data not serve as a ground truth for the validation of an analysis framework described here? Is there a particular reason why this or other type of 3D image data are not manually analyzed to check the accuracy of the computational framework?

*The article “Quantitative mRNA imaging throughout the entire Drosophila brain” by Long et al. describes a method to perform smFISH in the Drosophila brain with subcellular resolution. While RNA localization in cells can be observed in these data, subcellular localization is clearly not the main target of the study and the experimental methods have not been optimized to address this question (for instance, more markers would be needed to calculate localization features at the cellular level). Importantly, there is no ground truth associated to this article, in terms of sub-cellular localization. This is actually a general problem: the scientific community is certainly not lacking data, but it is in need for more **annotated data**. Regarding the difficulty of annotation of RNA localization data, we would like to refer to our answer to R3.1 above.*

On a different note, it is of course interesting to think about possible extensions of our method to make it applicable in the context of organs or organisms. Regarding this, we would like to refer to our answer R1.4.

I did not mean to say that the study, which I just cited by way of example, would already contain annotated data. I was wondering, whether there is a good reason not to take some of this kind of data and actually perform a manual annotation. The validation of your computational approach remains open to be quantitatively validated and, indeed, this requires some data with manual annotation.

R 3.3 As it is presented now, it appears to me that the authors simply did not have access to or did not team up with experimentalists to design their analysis framework. In fact, the fairly complex analysis framework could be much more streamlined by being based on ground truth data. In its current state, I wonder whether it will be ever useful to the scientific community; apart from the fact that most of the code seems to be written in the commercial software environment MATLAB that is not widely accessible in experimental labs. My impression is that the usefulness of this computational tool will be highly limited due to its complexity and design.

We deeply regret that our manuscript created this impression. As a matter of fact, more than half of the authors are experimental scientists that are familiar with smFISH and RNA localization for more than 15 years (see Basyuk et al., 2003; Bertrand et al., 1998; Fusco et al., 2003 by lead author E. Bertrand). The project has been designed in a collaborative effort to address the most pressing practical problem in High Content Screening for RNA localization today, and the data presented in this article has been generated by the authors.

[REDACTED]

I did not carefully check the composition of the research team before and I apologize for this. Now, as it is composed of experimentalists and theoreticians, they should together be able to agree on a ground truth data set. This is exactly the way we routinely do these kind of studies.

*Regarding MATLAB: while it is correct that **Matlab** is a commercial software, we disagree with the reviewer that this makes the usefulness of our approach questionable. Matlab is widely used for image processing and readily available in many labs or imaging platforms. Many published papers by the NATURE PUBLISHING GROUP are accompanied by Matlab source code. As an example, our software package FISH-quant to detect mRNAs in smFISH, is written in Matlab, and widely used in many research labs (66 citations since its publication in 2013; source: web of science).*

*Regarding the **complexity of the workflow**. We would like to point out that the application of our method with the proposed feature set and in an unsupervised setting is actually straightforward. We used the simulation framework to design and benchmark feature sets and the machine learning workflow, but the simulation is not required for an unsupervised analysis. Regarding the usefulness and justification of the simulation part, we would like to refer to answer R3.1.*

There is no doubt that the application of your approach will be limited by the use of commercial software, such as MATLAB. The fact that many published papers are accompanied by MATLAB source code does not change this argument at all. Providing citation numbers is again not an argument against this, unless you would compare with identical software packages that are provided open source.

Regarding the complexity of the workflow and its practical applicability, I would like to wait and see whether all issues have indeed been resolved by experts, who could not work with the previously provided version, as stated by Reviewer #2, comment R2.9.

Even then, I am missing a compact overview of all parameters that need to be set for the analysis.

R 3.4 If indeed it would not be possible to build and validate the analysis framework on ground truth data, then it does not seem appropriate to refer to the synthetically generated data in this study as being "realistic". It is at least not clear, what is meant by such a statement: How would unrealistic synthetic data look like? The authors need to explain their approach in a more rigorous way making it (sound?) less illogical.

First of all, as mentioned in R3.1, we can manually annotate images (see R3.7 below), but it does not seem to be a scalable approach in this context.

Manual annotation is meant to be done for a scalable approach. It is meant to be done to get an idea on the general performance of the computational approach.

A pragmatic, yet subjective definition of realistic images would be that in a side-by-side comparison, it is not immediately obvious which images are experimental and which are simulated. This was tested on experienced microscopists (e.g. FM, EB and MP), and it can be directly appreciated from the example images that we provide. In addition, we compared experimental and simulated data in terms of the extracted localization features. For many patterns we observe very similar feature distributions (See Supplementary Fig S5-9), indicating that the images are also realistic with respect to the extracted localization features.

I feel that the authors are arguing many ways without getting convinced to go the standard and straightforward way of validating their approach.

R 3.5 Even though the supplementary material is very extended and describes the approach in great detail, I missed a table where all parameters of the analysis are summarized. As a consequence, it is not clear to me how many parameters there are.

We now extended the helpfile describing the use of the program (see R2.8 above). We provide a visual overview of the different steps and the main required parameters in the user manual. We do not believe that a big table with all parameters will be of much use, however. The parameters concern different steps of the workflow and do not need to be jointly optimized or tweaked. For instance, some parameters are used for cell segmentation and RNA detection, other parameters determine how different features are calculated. Regarding simulation parameters, we would like to refer to R3.6.

The authors state that they “do not believe that a big table with all parameters will be of much use”. I disagree, because this would help to see what it takes to determine when really following their approach.

R 3.6 Moreover, the parameters that are used all need justification and where possible even a comment with regard to their sensitivity regarding the analysis itself. For example, it is not sufficient to just state "An mRNA is considered to be localized at the nuclear membrane if it is below a fixed distance (800 nm) from the membrane..." (page 6 of supplementary material and this is repeated later on, e.g. page 7). Unfortunately, no justification why the parameter value of 800 nm should be reasonable is given.

This question refers to parameters that are only used only for the simulations. We would like to emphasize again that the parameters used in the simulation have no impact on the features calculated on real data (see also comment R2.7). In the unsupervised setting (which is most likely the predominant application case), there is therefore no impact on the performance of the algorithm. If really extreme values were chosen for the simulation (in this example 0 nm or 20 μ m), they might be useless for benchmarking, because the resulting images would either be too easy to classify (e.g. a distance of 0 nm from the membrane with standard deviation of 0) or too different from the actual pattern they are supposed to represent (e.g. if the RNA were at a distance of 30% of the cell width, they could hardly be considered as localizing close to the membrane). If the parameters of the simulation are set in a reasonable range, there should not impact on the actual algorithm, only on the reported accuracy.

Second, the situation is different in the supervised setting, where we learn from the simulated data. Here, our guess was that the impact of most parameters would be minor. In the example mentioned above, our guess was that the classifier will still learn the relevant signatures, whether we choose 800nm or 1200nm, as the feature profile will vary little for these values as compared to feature vectors for different localization patterns.

It is important to stress that we do not intend to provide a perfect biophysical model for RNA localization that takes into account the biophysical processes inside the cell. The simulation rather plays the role of data augmentation in the machine learning literature, where we provide reasonable examples for the different classes. In other words: the question is not whether the true distance between the membrane and the RNAs localizing at the membrane is 800 nm or 1200 nm, but to find a method that distinguishes the class “close to the membrane” from the other patterns, such as “random localization” or “foci”.

Nevertheless, to illustrate this point further and to try to convince the reviewer, we simulated images with localization towards the cell membrane in 2D with different values of the specified distance (800nm as before and 1200nm). We then compared these simulations with other localization patterns (cell membrane in 3D, nuclear envelope, and random Fig R11, left). k-means clustering reveals that the 2 different simulations with a

localization towards the cell membrane 2D are grouped together: even if we impose $k=5$ (i.e. we force the algorithm to perform an additional split), the two classes corresponding to the different parameter settings are almost undistinguishable. If we choose $k=4$ (Figure R11, right), we see that the cells for both cell membrane 2D simulations are clustered together. We therefore conclude that the impact of this parameter is minor on the type of analysis we are performing.

Figure R11. *k*-means clustering of 5 different localization patterns as indicated on the *x*-axis. **(a)** Clustering with 5 classes shows that the simulations towards the cell membrane are often clustered together. **(b)** Clustering with 4 classes shows that the simulations towards the cell membrane in 2D are grouped together and separated from the other three classes.

I am thankful to the authors for this in-depth response. However, I am left with the feeling that the analysis that they needed to do in order to finally come to the conclusion “that the impact of this parameter is minor on the type of analysis we are performing” simply shows how unclear, in general, the impact of the many parameter values is that they would need to check to come to such a conclusion for a different data set.

R 3.7 In conclusion, the complexity of the computational framework that is largely based on commercial software is overwhelming making its usefulness highly questionable, all the more that it remains unclear how realistic the synthetic cell representation really is. This main issue could only be solved if real data would be manually analyzed and used as a ground truth.

To address this point, we performed a comparison of simulation versus manual annotations. We trained a Random Forest classifier on simulated data for 5 classes (Foci, cell extension, nuclear envelope in 2D, nuclear envelope in 3D, and random), and we applied it to experimental data. We then performed manual annotations on the experimental data. The confusion matrix between these two annotations is shown below in Fig R12. We observe that there is generally good agreement between the two methods. The two main cases of disagreement concern “foci” which are confused with “localization at the nuclear envelope in 2D” (which is due to foci in close proximity to the nuclear edge) and “cell extension” which are sometimes confused with “random” indicating the difficulty of assessing how significant an accumulation in the extension is. However, the overall agreement is quite good and comparable to performances used in morphological phenotyping (see for instance Neumann et al., 2010). We note however, that it cannot be concluded that the algorithm has an accuracy of 83%, as the manual annotations can be questioned as much as the results obtained by automatic classification.

Fig. R12. Comparison of automated classification with random forest and manual annotation. Cells were manually annotated and automatically classified by a classifier trained on simulated data. Shown is the confusion matrix between this manual annotation and automated classification.

I do not agree with the authors that “it cannot be concluded that the algorithm has an accuracy of 83%”, because their argument that the “manual annotation can be questioned” is meaningless. From my point of view, it is this type of thinking that causes the different views on their study.

Yes, I agree, manual annotation will not be perfect. But I do not agree with the implication that the authors seem to make, *i.e.* there would be the possibility that the algorithmic approach would effectively perform better than what they have found, because the manual annotation might be partly wrong. If we do think along such lines, we are starting to discuss about issues that – by definition – we will never be able to settle. From my point of view, one should never trust an automatic evaluation, if (i) it has not been quantitatively validated by performance measures on real data and (ii) it is said to be impossible to be validated because a reliable manual annotation would not be available.

It would be helpful, if the authors could show that our discrepancies are due to a misunderstanding on my side. Otherwise, I would like to insist on a quantitative validation of their computational approach by performance measures compared with a manual annotation of data.

If the authors still disagree, I suggest the editor to request a review of this manuscript from another reviewer.

Minor issues:

R 3.8 The authors should carefully check and correct their manuscript, including the supplementary material, for various typos and mistakes. For example:

- 1) Supplement, page 15, Fig S2-4 c: Label in the plot: not -> no
- 2) Supplement, page 19, last sentence of Section 2.5: There is one "and" too much in this sentence.
- 3) Supplement, page 21, top of page: The averaging formula should be presented as an equation (do not just give the right hand side).

We corrected these and other typos in the manuscript.

References

- Basyuk, E., Galli, T., Mougél, M., Blanchard, J.-M., Sitbon, M., and Bertrand, E. (2003). Retroviral genomic RNAs are transported to the plasma membrane by endosomal vesicles. *Dev. Cell* 5, 161–174.
- Battich, N., Stoeger, T., and Pelkmans, L. (2013). Image-based transcriptomics in thousands of single human cells at single-molecule resolution. *Nat. Methods* 10, 1127–1133.
- Bertrand, E., Chartrand, P., Schaefer, M., Shenoy, S.M., Singer, R.H., and Long, R.M. (1998). Localization of ASH1 mRNA particles in living yeast. *Mol. Cell* 2, 437–445.
- Cabili, M.N., Dunagin, M.C., McClanahan, P.D., Biaesch, A., Padovan-Merhar, O., Regev, A., Rinn, J.L., and Raj, A. (2015). Localization and abundance analysis of human lncRNAs at single-cell and single-molecule resolution. *Genome Biol.* 16, 20.
- Chen, K.H., Boettiger, A.N., Moffitt, J.R., Wang, S., and Zhuang, X. (2015). Spatially resolved, highly multiplexed RNA profiling in single cells. *Science* 348, 609–614.
- Fusco, D., Accornero, N., Lavoie, B., Shenoy, S.M., Blanchard, J.-M., Singer, R.H., and Bertrand, E. (2003). Single mRNA molecules demonstrate probabilistic movement in living mammalian cells. *Curr. Biol.* 13, 161–167.
- Held, M., Schmitz, M.H.A., Fischer, B., Walter, T., Neumann, B., Olma, M.H., Peter, M., Ellenberg, J., and Gerlich, D.W. (2010). CellCognition: time-resolved phenotype annotation in high-throughput live cell imaging. *Nat. Methods* 7, 747–754.
- Hubstenberger, A., Courel, M., Bénard, M., Souquere, S., Ernoult-Lange, M., Chouaib, R., Yi, Z., Morlot, J.-B., Munier, A., Fradet, M., et al. (2017). P-Body Purification Reveals the Condensation of Repressed mRNA Regulons. *Mol. Cell* 68, 144–157.e5.
- Jambor, H., Surendranath, V., Kalinka, A.T., Mejsstrik, P., Saalfeld, S., and Tomancak, P. (2015). Systematic imaging reveals features and changing localization of mRNAs in *Drosophila* development. *ELife* 4.
- Jones, T.R., Carpenter, A.E., Lamprecht, M.R., Moffat, J., Silver, S.J., Grenier, J.K., Castoreno, A.B., Eggert, U.S., Root, D.E., Golland, P., et al. (2009). Scoring diverse cellular morphologies in image-based screens with iterative feedback and machine learning. *Proc. Natl. Acad. Sci. U. S. A.* 106, 1826–1831.
- Lécuyer, E., Yoshida, H., Parthasarathy, N., Alm, C., Babak, T., Cerovina, T., Hughes, T.R., Tomancak, P., and Krause, H.M. (2007). Global analysis of mRNA localization reveals a prominent role in organizing cellular architecture and function. *Cell* 131, 174–187.
- Mili, S., Moissoglu, K., and Macara, I.G. (2008). Genome-wide screen reveals APC-associated RNAs enriched in cell protrusions. *Nature* 453, 115–119.
- Neumann, B., Walter, T., Hériché, J.-K., Bulkescher, J., Erfle, H., Conrad, C., Rogers, P., Poser, I., Held, M., Liebel, U., et al. (2010). Phenotypic profiling of the human genome by time-lapse microscopy reveals cell division genes. *Nature* 464, 721–727.
- Pichon, X., Bastide, A., Safieddine, A., Chouaib, R., Samacoits, A., Basyuk, E., Peter, M., Mueller, F., and Bertrand, E. (2016). Visualization of single endogenous polysomes reveals the dynamics of translation in live human cells. *J. Cell Biol.* 214, 769–781.
- Stoeger, T., Battich, N., Herrmann, M.D., Yakimovich, Y., and Pelkmans, L. (2015). Computer vision for image-based transcriptomics. *Methods* 77, 101–110.
- Wilk, R., Hu, J., Blotsky, D., and Krause, H.M. (2016). Diverse and pervasive subcellular distributions for both coding and long noncoding RNAs. *Genes Dev.* 30, 594–609.

Reviewers 1 and 2 asked for minor modifications. We have followed all of the suggestions, corrected the spelling mistakes and updated the tutorial.

Reviewer 3 had two concerns.

R3.1 Reviewer 3 asked for a comparison between the performance of the random forest classifier trained on manually annotated real images and the classifier trained on simulated data. The reviewer reasons that the accuracy of 83% in Fig. R12 does not reassure them of the sufficient similarity of the simulated images to real ones. Reviewer 3 also asks that to provide all possible performance measures (accuracy, precision, recall, specificity, F1 score).

*We performed the requested analysis and found that a random forest classifier trained on manually annotated real images and a random forest classifier trained on simulated data show nearly identical performance (**Supplementary Fig S5-11**, pages 50-51). These results show that the feature distributions of simulated and real data are close enough to allow to train a classifier with simulated data without a notable loss in performance.*

*We would like to point out that there are several use-cases of our simulation framework that go beyond replacing manual annotation of known patterns from already identified genes. One strength of our framework is that we can look for particular patterns **without** having identified example genes. In addition, our simulation framework can be used for validation and benchmarking of unsupervised approaches, which is a particularly important use-case in the field.*

R3.2 Reviewer 3 requested the addition of a table summarizing the parameters of the simulation workflow.

We now added this table to Supplementary Note 1 (**Table S1-1**, page 10).

REVIEWERS' COMMENTS:

Reviewer #3 (Remarks to the Author):

I am thankful to the authors for including the requested data into the manuscript. All my concerns have been addressed and I now recommend the manuscript of publication in Nat Commun.